# *Plasmodium falciparum* K13 mutations in Africa and Asia impact artemisinin resistance and parasite fitness

Barbara H Stokes[1], Satish K Dhingra[1], Kelly Rubiano[1], Sachel Mok[1], Judith Straimer[1], Nina F Gnädig[1], Ioanna Deni[1], Kyra A Schindler[1], Jade R Bath[1], Kurt E Ward[1,2], Josefine Striepen[1], Tomas Yeo[1], Leila S Ross[1], Eric Legrand[3], Frédéric Ariey[4], Clark H Cunningham[5], Issa M Souleymane[6], Adama Gansané[7], Romaric Nzoumbou-Boko[8], Claudette Ndayikunda[9], Abdunoor M Kabanywanyi[10], Aline Uwimana[11], Samuel J Smith[12], Olimatou Kolley[13], Mathieu Ndounga[14], Marian Warsame[15], Rithea Leang[16], François Nosten[17,18], Timothy JC Anderson[19], Philip J Rosenthal[20], Didier Ménard[3], David A Fidock[21]*

[1]Department of Microbiology and Immunology, Columbia University Irving Medical Center, New York, United States; [2]Department of Microbiology and Immunology, University of Otago, Dunedin, New Zealand; [3]Malaria Genetics and Resistance Unit, Institut Pasteur, INSERM U1201, CNRS ERL9195, Paris, France; [4]Institut Cochin, INSERM U1016, Université Paris Descartes, Paris, France; [5]Department of Genetics, University of North Carolina at Chapel Hill, Chapel Hill, United States; [6]Programme National de Lutte Contre le Paludisme au Tchad, Ndjamena, Chad; [7]Centre National de Recherche et de Formation sur le Paludisme, Ouagadougou, Burkina Faso; [8]Laboratoire de Parasitologie, Institut Pasteur de Bangui, Bangui, Central African Republic; [9]University Teaching Hospital of Kamenge, Bujumbura, Burundi; [10]Ifakara Health Institute, Dar es Salaam, United Republic of Tanzania; [11]Malaria and Other Parasitic Diseases Division, Rwanda Biomedical Centre, Kigali, Rwanda; [12]National Malaria Control Program, Freetown, Sierra Leone; [13]National Malaria Control Program, Banjul, Gambia; [14]Programme National de Lutte Contre le Paludisme, Brazzaville, Democratic Republic of the Congo; [15]School of Public Health and Community Medicine, University of Gothenburg, Gothenburg, Sweden; [16]National Center for Parasitology, Entomology & Malaria Control, Phnom Penh, Cambodia; [17]Shoklo Malaria Research Unit, Mahidol-Oxford Tropical Medicine Research Unit, Faculty of Tropical Medicine, Mahidol University, Mae Sot, Thailand; [18]Centre for Tropical Medicine and Global Health, Nuffield Department of Medicine, University of Oxford, Oxford, United Kingdom; [19]Texas Biomedical Research Institute, San Antonio, United States; [20]Department of Medicine, University of California, San Francisco, San Francisco, United States; [21]Division of Infectious Diseases, Department of Medicine, Columbia University Irving Medical Center, New York, United States

*For correspondence:
df2260@cumc.columbia.edu

Competing interests: The authors declare that no competing interests exist.

**Abstract** The emergence of mutant K13-mediated artemisinin (ART) resistance in *Plasmodium falciparum* malaria parasites has led to widespread treatment failures across Southeast Asia. In Africa, *K13*-propeller genotyping confirms the emergence of the R561H mutation in Rwanda and highlights the continuing dominance of wild-type K13 elsewhere. Using gene editing, we show that R561H, along with C580Y and M579I, confer elevated in vitro ART resistance in some African

strains, contrasting with minimal changes in ART susceptibility in others. C580Y and M579I cause substantial fitness costs, which may slow their dissemination in high-transmission settings, in contrast with R561H that in African 3D7 parasites is fitness neutral. In Cambodia, *K13* genotyping highlights the increasing spatio-temporal dominance of C580Y. Editing multiple K13 mutations into a panel of Southeast Asian strains reveals that only the R561H variant yields ART resistance comparable to C580Y. In Asian Dd2 parasites C580Y shows no fitness cost, in contrast with most other K13 mutations tested, including R561H. Editing of point mutations in *ferredoxin* or *mdr2*, earlier associated with resistance, has no impact on ART susceptibility or parasite fitness. These data underline the complex interplay between K13 mutations, parasite survival, growth and genetic background in contributing to the spread of ART resistance.

## Introduction

Despite recent advances in chemotherapeutics, diagnostics and vector control measures, malaria continues to exert a significant impact on human health (*Hanboonkunupakarn and White, 2020*). In 2019, cases were estimated at 229 million, resulting in 409,000 fatal outcomes, primarily in Sub-Saharan Africa as a result of *Plasmodium falciparum* infection (*WHO, 2020*). This situation is predicted to worsen as a result of the ongoing SARS-CoV-2 pandemic that has compromised malaria treatment and prevention measures (*Sherrard-Smith et al., 2020*). In the absence of an effective licensed malaria vaccine, control and elimination strategies are critically reliant on the continued clinical efficacy of first-line artemisinin-based combination therapies (ACTs) (*White et al., 2014*). These ACTs pair fast-acting artemisinin (ART) derivatives with partner drugs such as lumefantrine, amodiaquine, mefloquine, or piperaquine (PPQ). ART derivatives can reduce the biomass of drug-sensitive parasites by up to 10,000-fold within 48 hr (the duration of one intra-erythrocytic developmental cycle); however, these derivatives are rapidly metabolized in vivo. Longer-lasting, albeit slower-acting, partner drugs are co-administered to reduce the selective pressure for ART resistance and to clear residual parasitemias (*Eastman and Fidock, 2009*).

*P. falciparum* resistance to ART derivatives has now swept across Southeast (SE) Asia, having first emerged a decade ago in western Cambodia (*Dondorp et al., 2009*; *Noedl et al., 2009*; *Ariey et al., 2014*; *Imwong et al., 2020*). Clinically, ART resistance manifests as delayed clearance of circulating asexual blood stage parasites following treatment with an ACT, but does not result in treatment failure as long as the partner drug remains effective. The accepted threshold for resistance is a parasite clearance half-life (the time required for the peripheral blood parasite density to decrease by 50%) of >5.5 hr. Sensitive parasites are typically cleared in <2–3 hr (*WHO, 2019*). Resistance can also be evidenced as parasite-positive blood smears on day three post initiation of treatment. In vitro, ART resistance manifests as increased survival of tightly synchronized early ring-stage parasites (0–3 hr post invasion) exposed to a 6 hr pulse of 700 nM dihydroartemisinin (DHA, the active metabolite of all ARTs used clinically) in the ring-stage survival assay (RSA) (*Witkowski et al., 2013*; *Ariey et al., 2014*). Recently, ART-resistant strains have also acquired resistance to PPQ, which is widely used in SE Asia as a partner drug in combination with DHA (*Wicht et al., 2020*). Failure rates following DHA-PPQ treatment now exceed 50% in parts of Cambodia, Thailand, and Vietnam (*van der Pluijm et al., 2019*).

In vitro selections, supported by clinical epidemiological data, have demonstrated that ART resistance is primarily determined by mutations in the beta-propeller domain of the *P. falciparum* Kelch protein K13, also known as Kelch13 (*Ariey et al., 2014*; *Ashley et al., 2014*; *MalariaGEN Plasmodium falciparum Community Project, 2016*; *Ménard et al., 2016*; *Siddiqui et al., 2020*). Recent evidence suggests that these mutations result in reduced endocytosis of host-derived hemoglobin and thereby decrease the release of $Fe^{2+}$-heme that serves to activate ART, thus reducing the drug's potency (*Yang et al., 2019*; *Birnbaum et al., 2020*). Mutations in other genes including *ferredoxin (fd)* and *multidrug resistance protein 2 (mdr2)* have also been associated with ART resistance in K13 mutant parasites, leading to the suggestion that they may contribute to a multigenic basis of resistance and/or parasite fitness, or serve as genetic markers of founder populations (*Miotto et al., 2015*).

In SE Asia, the most prevalent K13 mutation is C580Y, which associates with delayed clearance in vivo (*Ariey et al., 2014*; *Ashley et al., 2014*; *MalariaGEN Plasmodium falciparum Community*

*Project, 2016*; *Ménard et al., 2016*; *Imwong et al., 2017*). This mutation also mediates ART resistance in vitro, as demonstrated by RSAs performed on gene-edited parasites (*Ghorbal et al., 2014*; *Straimer et al., 2015*; *Straimer et al., 2017*; *Mathieu et al., 2020*; *Uwimana et al., 2020*). Other studies have documented the emergence of nearly 200 other K13 mutations, both in SE Asia and in other malaria-endemic regions, including the Guiana Shield and the western Pacific (*MalariaGEN Plasmodium falciparum Community Project, 2016*; *Ménard et al., 2016*; *Das et al., 2019*; *WWARN K13 Genotype-Phenotype Study Group, 2019*; *Mathieu et al., 2020*; *Miotto et al., 2020*). Aside from C580Y, however, only a handful of other K13 mutations (N458Y, M476I, Y493H, R539T, I543T, and R561H) have been validated by gene-editing experiments as conferring ART resistance in vitro (*Straimer et al., 2015*; *Siddiqui et al., 2020*). Nonetheless, multiple mutations in this gene have been associated with the clinical delayed clearance phenotype and have been proposed as candidate markers of ART resistance (*WWARN K13 Genotype-Phenotype Study Group, 2019*; *WHO, 2019*).

Here, we define the role of a panel of K13 mutations identified in patient isolates, and address the key question of whether these mutations can confer resistance in African and Asian strains. We include the K13 R561H mutation, first associated with delayed parasite clearance in SE Asia (*Ashley et al., 2014*; *Phyo et al., 2016*), and very recently identified at up to 13% prevalence in certain districts in Rwanda (*Uwimana et al., 2020*; *Bergmann et al., 2021*; *Uwimana et al., 2021*). We also assess the impact of the parasite genetic background on in vitro phenotypes, including mutations in *ferredoxin* and *mdr2* that were earlier associated with resistance (*Miotto et al., 2015*). Our results show that K13 mutations can impart ART resistance across multiple Asian and African strains, at levels that vary widely depending on the mutation and the parasite genetic background. Compared with K13 mutant Asian parasites, we observed stronger in vitro fitness costs in most *K13* edited African strains, which might predict a slower dissemination of ART resistance in high-transmission African settings. Nonetheless, our data highlight the threat of the R561H mutation emerging in Rwanda, which confers elevated RSA survival and a minimal fitness cost in African 3D7 parasites.

## Results

### Non-synonymous K13 mutations are present at low frequencies in Africa

To examine the status of K13 mutations across Africa, we analyzed *K13* beta-propeller domain sequences in 3257 isolates from eleven malaria-endemic African countries, including The Gambia, Sierra Leone, and Burkina Faso in West Africa; Chad, Central African Republic, Republic of the Congo, and Equatorial Guinea in Central Africa; and Burundi, Tanzania, Rwanda, and Somalia in East Africa. Samples were collected between 2011 and 2019, with most countries sampled across multiple years. A total of 1038 (32%) samples originated from The Gambia, Republic of the Congo, or Burundi, and have not been previously reported, whereas the remaining samples, including those from Rwanda, have been published (*Figure 1—source data 1*; *Supplementary file 1*).

Of all samples, 98% (3179) were K13 wild-type, that is, they matched the 3D7 (African) reference sequence or harbored a synonymous (non-coding) mutation. For individual countries, the percentage of K13 wild-type samples ranged from 95% to 100% (*Figure 1*; *Figure 1—source data 1*). In total, we identified 35 unique non-synonymous mutations in K13. Of these, only two have been validated as resistance mediators in the SE Asian Dd2 strain: the M476I mutation initially identified from long-term ART selection studies, and the R561H mutation observed in SE Asia and now in Rwanda (*Ariey et al., 2014*; *Straimer et al., 2015*; *Uwimana et al., 2020*).

Of the 35 non-synonymous mutations, only two were present in ≥ten samples: R561H (n=20, found only in Rwanda, sampled from 2012 to 2015; *Uwimana et al., 2020*), and A578S (n=10; observed in four African countries across multiple years). Previously, A578S was shown not to confer in vitro resistance in Dd2 (*Ménard et al., 2016*). In the set of 927 genotyped Rwandan isolates, R561H accounted for 44% of mutant samples and 2% of all samples (*Figure 1* inset).

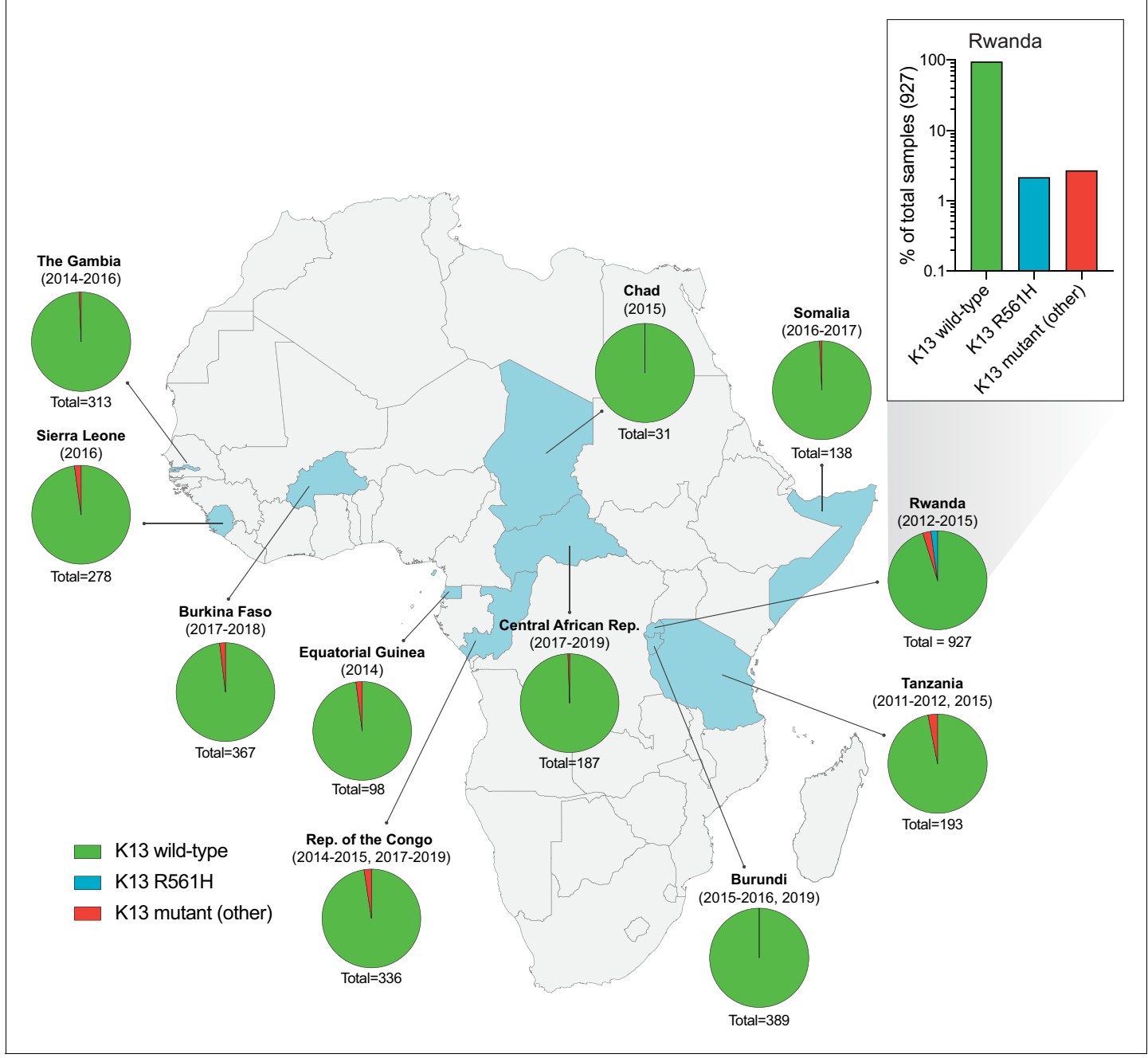

**Figure 1.** Frequency and distribution of *K13* alleles in eleven African countries. Map of Africa with pie charts representing the proportions of sequenced samples per country that harbor the K13 wild-type sequence (3D7 reference), the R561H variant (the most commonly identified mutation, unique to Rwanda; see inset), or another less frequent non-synonymous K13 mutation. Sample sizes and years of sample collection are indicated. Mutations and numbers of African samples sequenced per country, and prior citations as appropriate, are listed in *Figure 1—source data 1*.
The online version of this article includes the following source data for figure 1:

**Source data 1.** Distribution of *K13* alleles over time in African countries (2011–2019).

## K13 R561H, M579I, and C580Y mutations can confer in vitro artemisinin resistance in African parasites

To test whether the K13 R561H mutation could mediate ART resistance in African strains, we developed a CRISPR/Cas9-mediated *K13* editing strategy (*Supplementary file 2*) to introduce this mutation into 3D7 and F32 parasites. On the basis of whole-genome sequence analysis of African

isolates, 3D7 was recently shown to segregate phylogenetically with parasites from Rwanda (*Uwimana et al., 2020*). F32 was derived from an isolate from Tanzania (*Witkowski et al., 2010*). We also tested the C580Y mutation that predominates in SE Asia, as well as the M579I mutation earlier identified in a *P. falciparum*-infected migrant worker in Equatorial Guinea who displayed delayed parasite clearance following ACT treatment (*Lu et al., 2017*). The positions of these residues are highlighted in the K13 beta-propeller domain structure shown in *Supplementary file 3*. For 3D7, F32 and other lines used for this study, geographic origins and genotypes at drug resistance loci are described in *Table 1* and *Supplementary file 4*. All parental lines were cloned by limiting dilution prior to transfection. Edited parasites were identified by PCR and Sanger sequencing, and cloned. These and other edited parasite lines used herein are described in *Supplementary file 5*.

RSAs, used to measure in vitro ART susceptibility, revealed a wide range of mean survival values for K13 mutant lines. For 3D7 parasites, the highest RSA survival rates were observed with 3D7$^{R561H}$ parasites, which averaged 6.6% RSA survival. For the 3D7$^{M579I}$ and 3D7$^{C580Y}$ lines, mean RSA survival rates were both 4.8%, a three- to fourfold increase relative to the 3D7$^{WT}$ line. No elevated RSA survival was seen in a 3D7 control line (3D7$^{ctrl}$) that expresses only the silent shield mutations used at the guide RNA (gRNA) cut site (*Figure 2A*; *Figure 2—source data 1*). Western blots performed on tightly synchronized ring-stage parasites revealed an ~30% reduction in K13 protein expression levels in these three K13 mutant lines relative to the parental 3D7$^{WT}$ line (*Figure 2—figure supplement 1*; *Figure 2—figure supplement 1—source data 1*).

Interestingly, for F32 parasites, the introduction of K13 mutations yielded no significant increases in RSA survival, with survival rates in the range of 0.3% to 0.5% for lines expressing R561H, M579I, C580Y, or wild-type K13. (*Figure 2B*). Previously we reported that introduction of M476I into F32 parasites resulted in a modest gain of resistance (mean survival of 1.7%), while this same mutation conferred RSA survival levels of ~10% in edited Dd2 parasites (*Straimer et al., 2015*). These data suggest that while K13 mutations differ substantially in their impact on ART susceptibility, there is an equally notable contribution of the parasite genetic background.

We next introduced M579I and C580Y into the cloned Ugandan isolates UG659 and UG815. Editing of both mutations into UG659 yielded moderate RSA survival rates (means of 6.3% and 4.7% for UG659$^{M579I}$ or UG659$^{C580Y}$ respectively, vs 1.0% for UG659$^{WT}$; *Figure 2C*). These values resembled our results for 3D7. Strikingly, introducing K13 M579I or C580Y into UG815 yielded the highest rates

**Table 1.** *Plasmodium falciparum* lines employed herein.

| Parasite | Origin | Year | *K13* | Resistance |
|---|---|---|---|---|
| 3D7$^{WT}$ | Africa | 1981 | WT | – |
| F32$^{WT}$ | Tanzania | 1982 | WT | – |
| UG659$^{WT}$ | Uganda | 2007 | WT | CQ, SP |
| UG815$^{WT}$ | Uganda | 2008 | WT | CQ, SP |
| Dd2$^{WT}$ | Indochina | 1980 | WT | CQ, MFQ, SP |
| Cam3.II$^{WT}$ | Cambodia | 2010 | WT | CQ, SP |
| CamWT$^{C580Y}$ | Cambodia | 2010 | C580Y | ART, CQ, SP |
| RF7$^{C580Y}$ | Cambodia | 2012 | C580Y | ART, CQ, PPQ, SP |
| Thai1$^{WT}$ | Thailand | 2003 | WT | CQ, SP |
| Thai2$^{WT}$ | Thailand | 2004 | WT | CQ, MFQ, SP |
| Thai3$^{WT}$ | Thailand | 2003 | WT | CQ, SP |
| Thai4$^{WT}$ | Thailand | 2003 | WT | CQ, SP |
| Thai5$^{WT}$ | Thailand | 2011 | WT | CQ, SP |
| Thai6$^{E252Q}$ | Thailand | 2008 | E252Q | ART (low), CQ, MFQ, SP |
| Thai7$^{E252Q}$ | Thailand | 2010 | E252Q | ART (low), CQ, MFQ, SP |

Parasite superscripts refer to the K13 sequence.

ART, artemisinin; CQ, chloroquine; MFQ, mefloquine; PPQ, piperaquine; SP, sulfadoxine/pyrimethamine; WT, wild type.

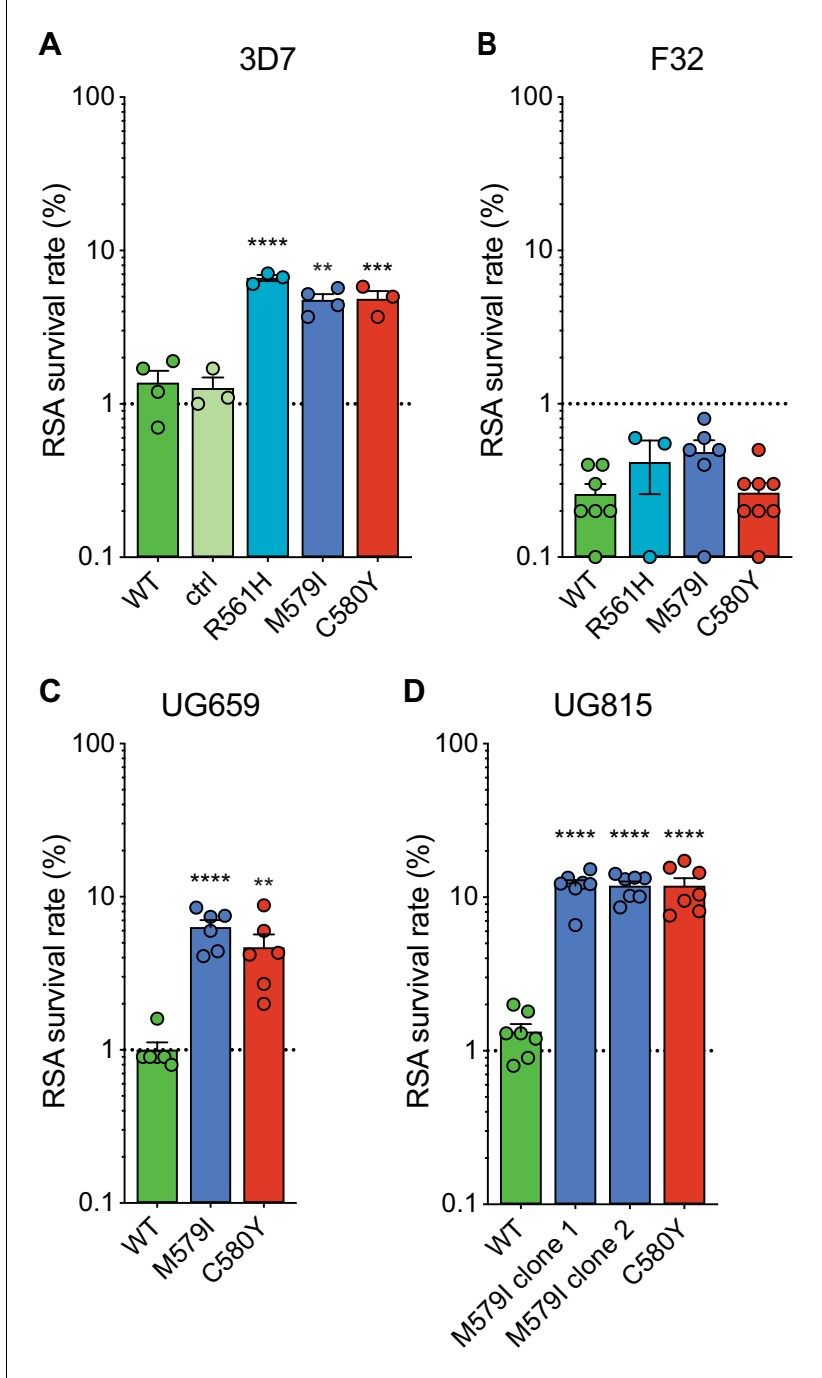

**Figure 2.** Gene-edited mutant *K13* African parasites display variable levels of RSA survival. (**A–D**) RSA survival rates for (**A**) 3D7 (Africa), (**B**) F32 (Tanzania), (**C**) UG659 (Uganda), or (**D**) UG815 (Uganda) K13 wild-type parental lines and CRISPR/Cas9-edited K13 R561H, M579I, or C580Y mutant clones. Unedited parental lines are described in *Table 1* and *Supplementary file 4*. For 3D7, we also included a K13 wild-type control (ctrl) line harboring silent shield mutations at the *K13* gRNA cut site. Results show the percentage of early ring-stage parasites (0–3 hr post invasion) that survived a 6 hr pulse of 700 nM DHA, relative to DMSO-treated parasites assayed in parallel. Percent survival values are shown as means ± SEM (detailed in *Figure 2—source data 1*). Results were obtained from three to eight independent experiments, each performed in duplicate. p Values were determined by unpaired *t* tests and were calculated for K13 mutant lines relative to their isogenic wild-type lines. ** p<0.01; *** p<0.001; **** p<0.0001.

The online version of this article includes the following source data and figure supplement(s) for figure 2:

*Figure 2 continued on next page*

*Figure 2 continued*

**Source data 1.** Ring-stage survival (RSA) assay data for *K13* edited African parasites and controls.
**Figure supplement 1.** K13 mutations result in reduced K13 protein levels in African 3D7 parasites.
**Figure supplement 1—source data 1.** Raw figure files for K13 Western blots performed on 3D7 parasites.

of in vitro resistance, with mean survival levels reaching ~12% in both UG815^M579I and UG815^C580Y. These results were confirmed in a second independent clone of UG815^M579I (*Figure 2D*). M579I and C580Y also conferred equivalent levels of resistance in edited Dd2 parasites (RSA survival rates of 4.0% and 4.7%, respectively; *Figure 2—source data 1*). These data show that mutant K13-mediated ART resistance in African parasites can be achieved in some strains at levels comparable to or above those seen in SE Asian parasites.

## K13 C580Y, M579I, and R561H mutations are associated with variable in vitro fitness costs in African parasites

To examine the relationship between resistance and fitness in African parasites harboring K13 mutations, we developed an in vitro fitness assay that uses quantitative real-time PCR (qPCR) for allelic discrimination. Assays were conducted by pairing K13 wild-type lines (3D7, F32, UG659, and UG815) with their isogenic *K13* edited R561H, M579I, or C580Y counterparts.

Assays were initiated with tightly synchronized trophozoites that were mixed in 1:1 ratios of wild-type to mutant isogenic parasites, and cultures were maintained over a period of 36 days (~18 generations of asexual blood stage growth). Cultures were sampled every four days for genomic DNA (gDNA) preparation and qPCR analysis. TaqMan probes specific to the *K13* wild-type or mutant (R561H, M579I, or C580Y) alleles were used to quantify the proportion of each allele.

Results showed that the K13 M579I and C580Y mutations each conferred significant fitness defects in most strains tested, with the proportions of K13 mutant lines declining over time. For both mutations, the largest reductions were observed with edited 3D7 or UG815 parasites. In contrast, these mutations exerted a minimal impact on fitness in UG659. For R561H, we observed no impact on fitness in 3D7 parasites, although in F32 this mutation exerted a fitness defect similar to M579I and C580Y (*Figure 3A–D*; *Figure 3—source data 1*). From these data, we calculated the fitness cost, which represents the percent reduction in growth rate per 48 hr generation of a test line compared to its wild-type isogenic comparator. These costs ranged from <1% to 12% per generation across mutations and lines, with the lowest costs observed in 3D7^R561H and UG659^C580Y parasites, and the greatest costs observed in the *K13* edited UG815 lines (*Figure 3E*). Comparing data across these four African strains revealed that high RSA survival rates were generally accompanied by high fitness costs, and conversely that low fitness costs were associated with low survival rates. Exceptions were the 3D7^R561H and UG659^C580Y lines that showed moderate resistance with little to no apparent fitness costs (*Figure 3F*).

## The K13 C580Y mutation has swept rapidly across Cambodia, displacing other K13 variants

We next examined the spatio-temporal distribution of *K13* alleles in Cambodia, the epicenter of ART resistance in SE Asia. In total, we analyzed *K13* propeller domain sequences from 3327 parasite isolates collected from fourteen Cambodian provinces in the western, northern, eastern, and southern regions (*Figure 4—figure supplement 1*). Samples were collected between 2001 and 2017, except for the southern region where sample collection was initiated in 2010. A total of 1412 samples (42%) were obtained and sequenced during the period from 2015–2017 and have not previously been published. Earlier samples were reported in *Ariey et al., 2014*; *Ménard et al., 2016*. In sum, 19 nonsynonymous polymorphisms in *K13* were identified across all regions and years. Of these, only three were present in >10 samples: Y493H (n=83), R539T (n=87), and C580Y (n=1915). Each of these mutations was previously shown to confer ART resistance in vitro (*Straimer et al., 2015*). Rarer mutations included A418V, I543T, P553L, R561H, P574L, and D584V (*Figure 4*; *Figure 4—source data 1*).

This analysis revealed a significant proportion of *K13* wild-type parasites in the early 2000s, particularly in northern and eastern Cambodia, where 96% of isolates in 2001–2002 were wild type

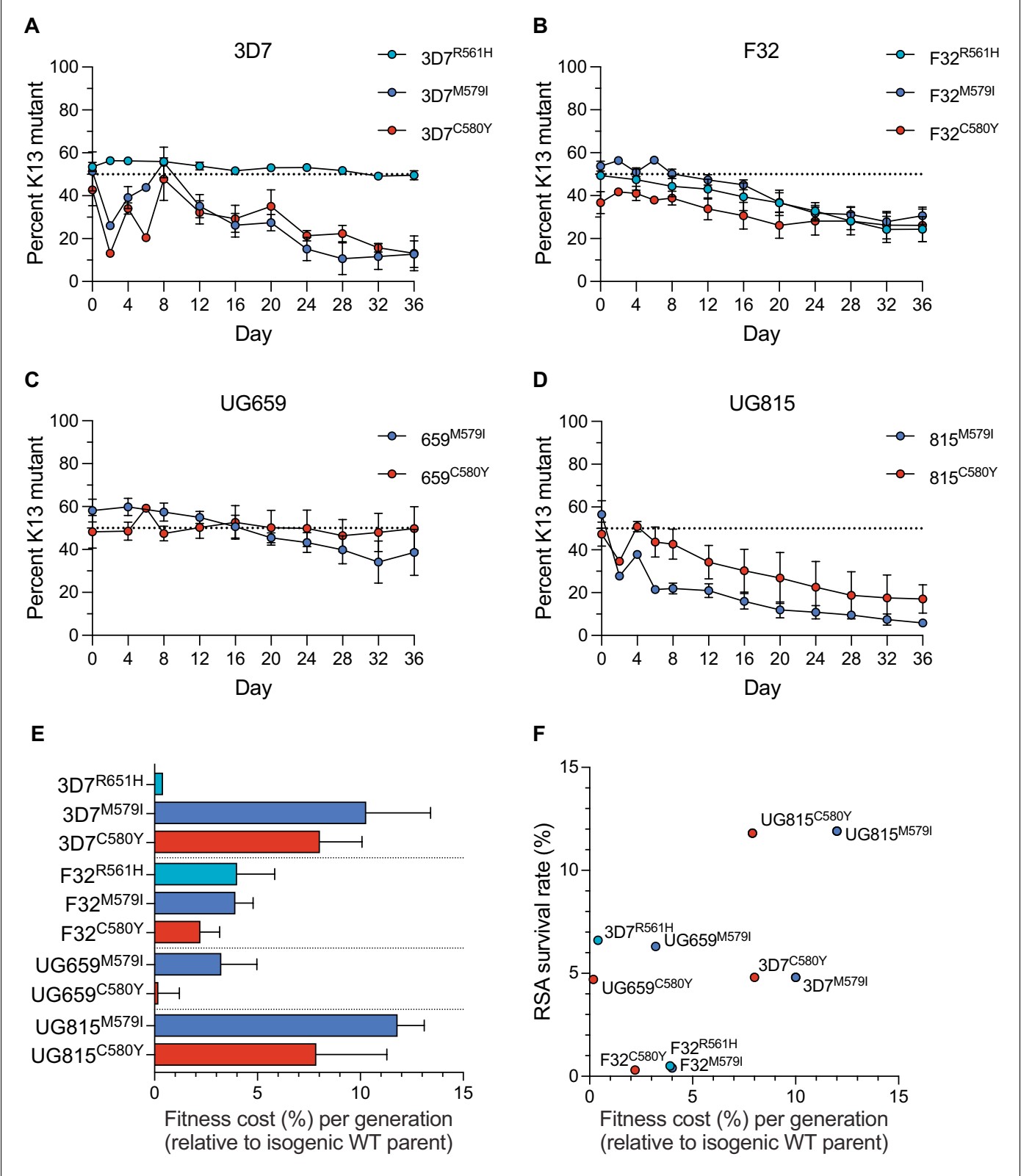

**Figure 3.** K13 mutations cause differential impacts on in vitro growth rates across gene-edited African strains. (A–D) Percentages of mutant alleles relative to the wild-type allele over time in (A) 3D7, (B) F32, (C) UG659, and (D) UG815 parasite cultures in which K13 mutant clones were co-cultured at 1:1 starting ratios with isogenic K13 wild-type controls over a period of 36 days. Results, shown as means ± SEM, were obtained from two to five independent experiments, each performed in duplicate. Values are provided in *Figure 3—source data 1*. (E) The percent reduction in growth rate per
*Figure 3 continued on next page*

*Figure 3 continued*

48 hr generation, termed the fitness cost, is presented as mean ± SEM for each mutant line relative to its isogenic wild-type comparator. (F) Fitness costs for mutant lines and isogenic wild-type comparators plotted relative to RSA survival values for the same lines.

The online version of this article includes the following source data for figure 3:

**Source data 1.** Fitness assay data for *K13* edited African parasite lines and controls.

(*Figure 4*). In western Cambodia, where ART resistance first emerged (*Dondorp et al., 2009*; *Noedl et al., 2009*), the wild-type allele percentage in 2001–2002 had already fallen to 56%. This is striking given that delayed parasite clearance following ACT or artesunate treatment was first documented in 2008–2009 (*Noedl et al., 2008*; *Noedl et al., 2009*).

In all four regions, the frequency of the wild-type allele declined substantially over time and the diversity of mutant alleles contracted, with nearly all wild-type and non-K13 C580Y mutant parasites being replaced by parasites harboring the C580Y mutation (*Figure 4*). This effect was particularly pronounced in the western and the southern regions, where the prevalence of C580Y in 2016–2017 effectively attained 100%, increasing from 22% and 58% in the initial sample sets, respectively (*Figure 4A,D*). In northern and eastern Cambodia, C580Y also outcompeted all other mutant alleles; however, 19–25% of parasites remained K13 wild type in 2016–2017 (*Figure 4B,C*). These data show rapid dissemination of K13 C580Y across Cambodia.

## SE Asian K13 mutations associated with delayed parasite clearance differ substantially in their ability to confer artemisinin resistance in vitro

Given that most K13 polymorphisms present in the field have not previously been characterized in vitro, we selected a set of mutations to test by gene editing, namely E252Q, F446I, P553L, R561H, and P574L. The positions of these residues are highlighted in *Supplementary file 3*. F446I is the predominant mutation in Myanmar (*Imwong et al., 2020*). P553L, R561H, and P574L have each been shown to have multiple independent origins throughout SE Asia (*Ménard et al., 2016*) and were identified at low frequencies in our sequencing study in Cambodia (*Figure 4*). Lastly, the E252Q mutation was formerly prevalent on the Thai-Myanmar border, and, despite its location upstream of the beta-propeller domain, has been associated with delayed parasite clearance in vivo (*Anderson et al., 2017*; *Cerqueira et al., 2017*; *WWARN K13 Genotype-Phenotype Study Group, 2019*).

Zinc-finger nuclease- or CRISPR/Cas9-based gene-edited lines expressing K13 E252Q, F446I, P553L, R561H, or P574L were generated in Dd2 or Cam3.II lines expressing wild-type K13 (Dd2[WT] or Cam3.II[WT]) and recombinant parasites were cloned. Early ring-stage parasites were then assayed for their ART susceptibility using the RSA. For comparison, we included published Dd2 and Cam3.II lines expressing either K13 C580Y (Dd2[C580Y] and Cam3.II[C580Y]) or R539T (Dd2[R539T] and the original parental line Cam3.II[R539T]) (*Straimer et al., 2015*), as well as control lines expressing only the guide-specific silent shield mutations (Dd2[ctrl] and Cam3.II[ctrl]).

Both the P553L and R561H mutations yielded mean RSA survival rates comparable to C580Y (4.6% or 4.3% RSA survival for Dd2[P553L] or Dd2[R561H], respectively, vs 4.7% for Dd2[C580Y]; *Figure 5A*; *Figure 5—source data 1*). F446I and P574L showed only modest increases in survival relative to the wild-type parental line (2.0% and 2.1% for Dd2[F446I] and Dd2[P574L], respectively, vs 0.6% for Dd2[WT]). No change in RSA survival relative to Dd2[WT] was observed for the Dd2[E252Q] line. The resistant benchmark Dd2[R539T] showed a mean RSA survival level of 20.0%, consistent with earlier reports of this mutation conferring high-grade ART resistance in vitro (*Straimer et al., 2015*; *Straimer et al., 2017*).

In contrast to Dd2, editing of the F446I, P553L, and P574L mutations into Cambodian Cam3.II[WT] parasites did not result in statistically significant increases in survival rates relative to the K13 wild-type line, in part because the background survival rate of Cam3.II[WT] was higher than for Dd2[WT]. All survival rates were <3%, contrasting with the Cam3.II[R539T] parental strain that expresses the R539T mutation (20.4% mean survival; *Figure 5B*; *Figure 5—source data 1*). The E252Q mutation did not result in elevated RSA survival in the Cam3.II background, a result also observed with Dd2. Nonetheless, ART resistance was apparent upon introduction of the R561H mutation into Cam3.II[WT]

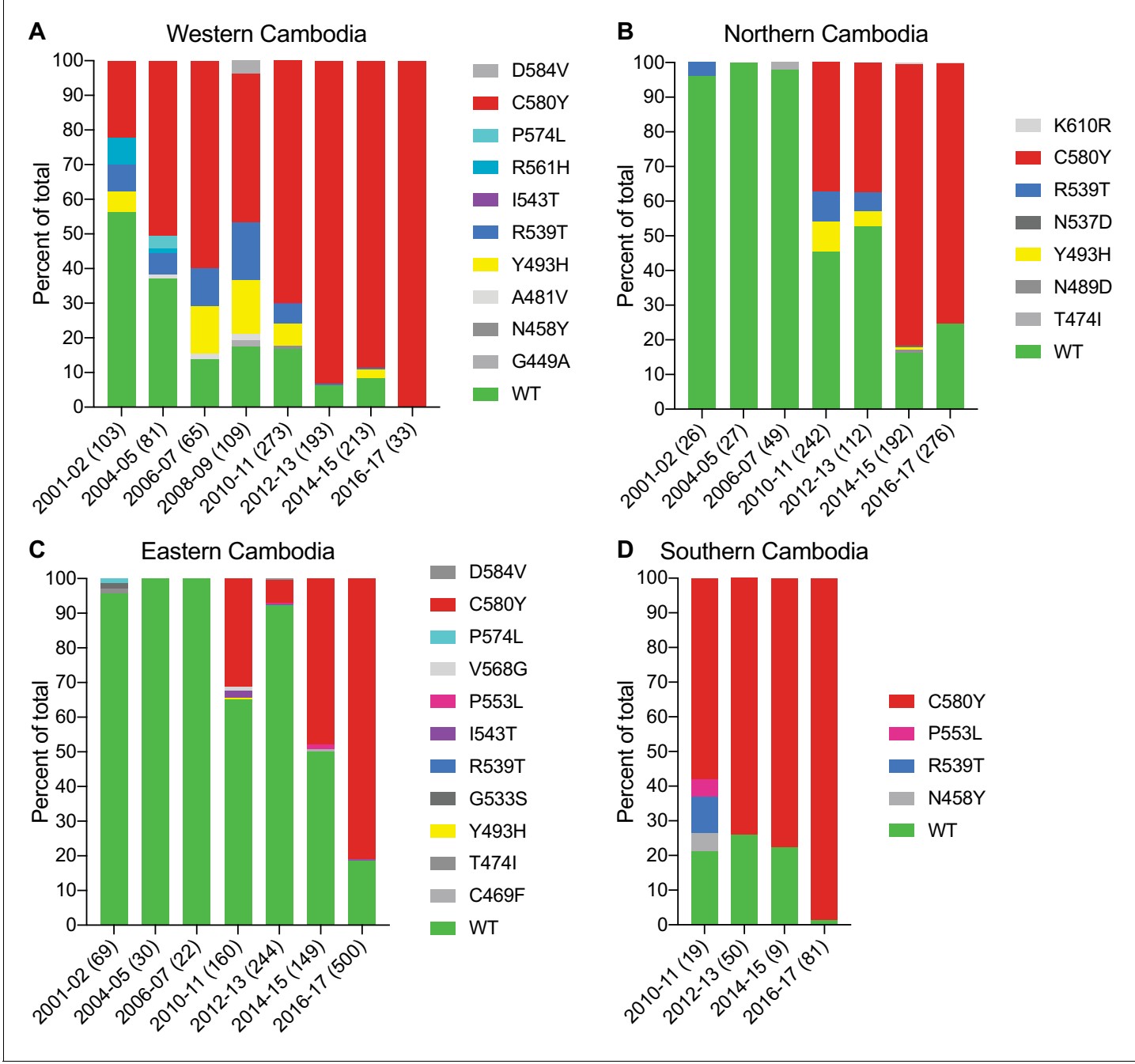

**Figure 4.** The *K13* C580Y allele has progressively outcompeted all other alleles in Cambodia. (A–D) Stacked bar charts representing the percentages of sequenced samples expressing the *K13* wild-type allele or individual variants, calculated based on the total number of samples (listed in parentheses) for a given period. Sample collection was segregated into four regions in Cambodia (detailed in *Figure 4—figure supplement 1*). All *K13* mutant samples harbored a single non-synonymous nucleotide polymorphism. Mutations and numbers of Cambodian samples sequenced per region/year, including prior citations as appropriate, are listed in *Figure 4—source data 1*.

The online version of this article includes the following source data and figure supplement(s) for figure 4:

**Source data 1.** Distribution of *K13* alleles over time in Cambodia (2001–2017).

**Figure supplement 1.** Regions of sample collection in Cambodia for *K13* sequencing.

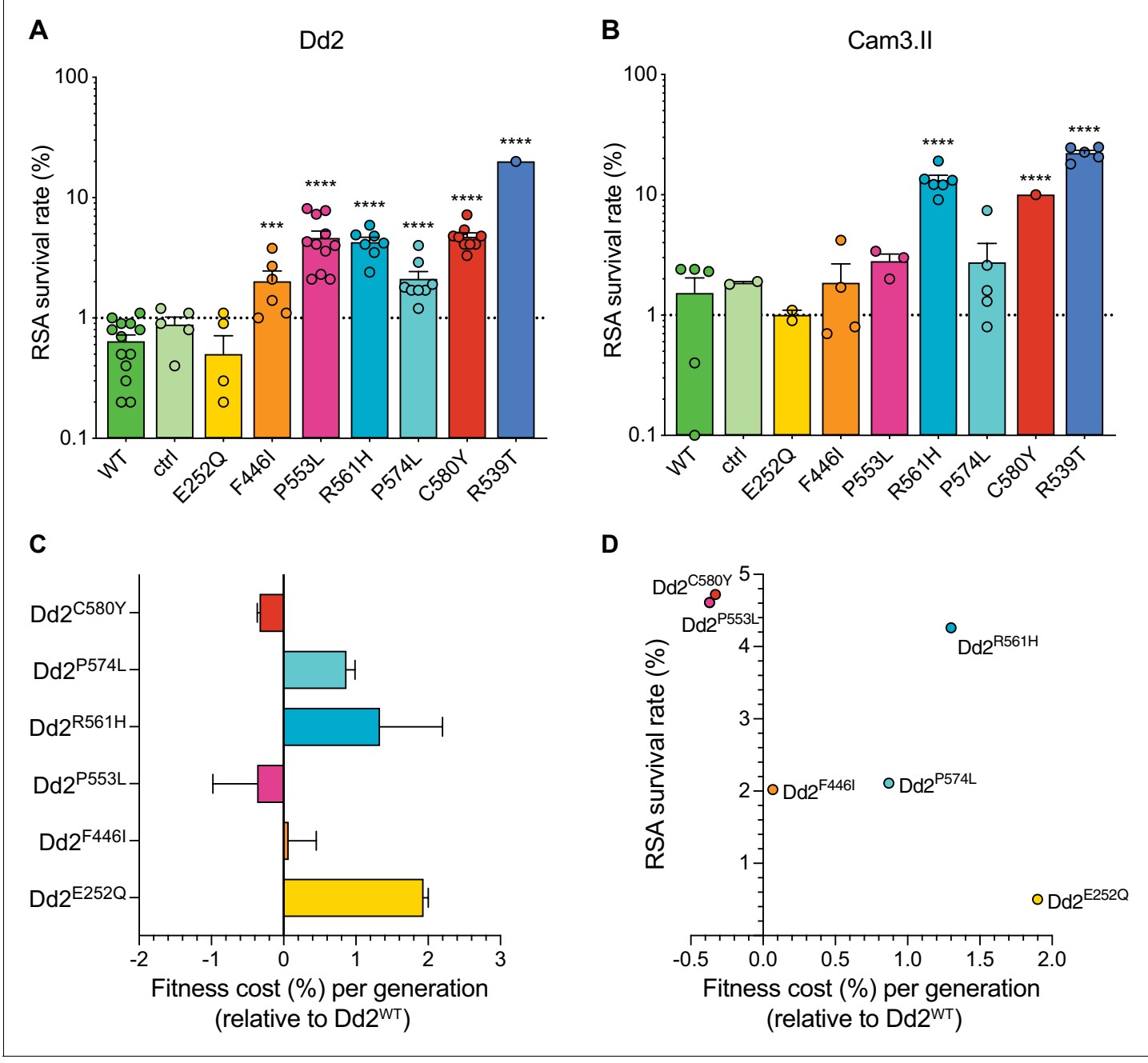

**Figure 5.** Southeast Asian K13 mutations yield elevated RSA survival and minor impacts on in vitro growth in gene-edited parasite lines. (A, B) RSA survival rates for Dd2 (Indochina) and Cam3.II (Cambodia) *P. falciparum* parasites expressing wild-type or mutant K13. Gene-edited parasites were generated using CRISPR/Cas9 or zinc-finger nucleases. Control (ctrl) lines express silent shield mutations at the *K13* gRNA cut site. Parental lines are described in *Table 1* and *Supplementary file 4*. Results show the percentages of early ring-stage parasites (0–3 hr post invasion) that survived a 6 hr pulse of 700 nM DHA, relative to DMSO-treated parasites processed in parallel. Percent survival values are shown as means ± SEM (detailed in *Figure 5—source data 1*). Results were obtained from three to thirteen independent experiments, each performed in duplicate. p Values were determined by unpaired *t* tests and were calculated for mutant lines relative to the isogenic line expressing wild-type *K13*. *** p<0.001; **** p<0.0001. (C) Percent reductions in growth rate per 48 hr generation, expressed as fitness costs, for Dd2 mutant lines relative to the Dd2^WT line. Fitness costs were determined by co-culturing the Dd2^eGFP reporter line with either the Dd2 K13 wild-type parental line (Dd2^WT) or gene-edited K13 mutant lines. Co-cultures were maintained for 20 days and percentages of eGFP+ parasites were determined by flow cytometry (see *Figure 5—source data 2* and *Figure 5—figure supplement 1*). Fitness costs were initially calculated relative to the Dd2^eGFP reporter line (*Figure 5—figure supplement 1*) and then normalized to the Dd2^WT line. Mean ± SEM values were obtained from three independent experiments, each performed in triplicate. (D) Fitness costs for K13 mutant lines relative to the Dd2^WT line plotted against their corresponding RSA survival values.

*Figure 5 continued on next page*

*Figure 5 continued*

The online version of this article includes the following source data and figure supplement(s) for figure 5:

**Source data 1.** Ring-stage survival (RSA) assay data for *K13* edited SE Asian parasites and controls (Dd2 and Cam3.II strains).
**Source data 2.** Fitness assay data (percent eGFP+ parasites) for *K13* edited Dd2 parasites and parental control.
**Figure supplement 1.** Southeast Asian K13 mutations result in minor in vitro growth defects in Dd2 parasites, with the exception of the C580Y and P553L mutations.

parasites, with mean survival rates for Cam3.II$^{R561H}$ exceeding those for the Cam3.II$^{C580Y}$ line (13.2% vs 10.0%, respectively). No elevated survival was seen in the Cam3.II$^{ctrl}$ line expressing only the silent shield mutations used at the gRNA cut site.

## SE Asian K13 mutations do not impart a significant fitness impact on Dd2 parasites

Prior studies with isogenic gene-edited SE Asian lines have shown that certain K13 mutations can exert fitness costs, as demonstrated by reduced intra-erythrocytic asexual blood stage parasite growth (*Straimer et al., 2017*; *Nair et al., 2018*). To determine the fitness impact of the K13 mutations described above, we used an eGFP-based parasite competitive growth assay (*Ross et al., 2018*). For these experiments, Dd2$^{E252Q}$, Dd2$^{F446I}$, Dd2$^{P553L}$, Dd2$^{R561H}$, or Dd2$^{P574L}$ parasites were co-cultured with a K13 wild-type eGFP-positive (eGFP$^+$) Dd2 reporter line at starting ratios of 1:1. Proportions of eGFP$^+$ parasites were then assessed every two days by flow cytometry. As controls, we also included the Dd2$^{WT}$, Dd2$^{bsm}$, and Dd2$^{C580Y}$ lines. These data provided evidence of a minimal growth impact with the F446I, P553L, and C580Y mutations. In contrast, E252Q, R561H, and P574L resulted in greater fitness costs when compared to Dd2$^{WT}$ parasites (*Figure 5C*; *Figure 5—figure supplement 1*; *Figure 5—source data 2*). Both the C580Y and P553L mutations yielded elevated RSA survival and minimal fitness costs in the Dd2 strain, providing optimal traits for dissemination (*Figure 5D*). We note that all fitness costs in *K13* edited Dd2 parasites were considerably lower than those observed in the majority of the *K13* edited African lines described above (*Figure 3*).

## Strain-dependent genetic background differences significantly impact RSA survival rates in culture-adapted Thai isolates

Given the earlier abundance of the R561H and E252Q alleles in border regions of Thailand and Myanmar, we next tested the impact of introducing these mutations into five K13 wild-type Thai isolates (Thai1-5). For comparison, we also edited C580Y into several of these same isolates. These studies revealed a major contribution of the parasite genetic background in dictating the level of mutant K13-mediated ART resistance, as exemplified by the C580Y mutation, which yielded mean survival rates ranging from 2.1% to 15.4% in edited parasites. Trends observed for individual mutations were maintained across strains, with the R561H mutation consistently yielding moderate to high in vitro resistance, at or above the level of C580Y. Consistent with our Dd2 results, E252Q edited parasites did not display significant increases in survival rates relative to isogenic K13 wild-type lines (*Figure 6A–E*; *Figure 6—source data 1*).

We also profiled two unedited culture-adapted Thai isolates (Thai6$^{E252Q}$ and Thai7$^{E252Q}$), which express the K13 E252Q mutation that occurs upstream of the propeller domain. Notably, both lines exhibited mean RSA survival rates above the 1% threshold for ART sensitivity (2.7% for Thai6$^{E252Q}$ and 5.1% for Thai7$^{E252Q}$; *Figure 6F*). These data suggest that additional genetic factors present in these two Thai isolates are required for E252Q to manifest ART resistance.

## Mutations in *P. falciparum multidrug resistance protein* 2 and *ferredoxin* do not modulate resistance to artemisinin or parasite fitness in vitro

In a prior genome-wide association study of SE Asian parasites, K13-mediated ART resistance was associated with the D193Y and T484I mutations in the *ferredoxin* (*fd*) and *multidrug resistance protein 2* (*mdr2*) genes, respectively (*Miotto et al., 2015*). To directly test the impact of these mutations in parasite resistance and fitness, we applied CRISPR/Cas9 editing (*Supplementary file 6*) to revert the fd D193Y and mdr2 T484I mutations to their wild-type sequences. These experiments were

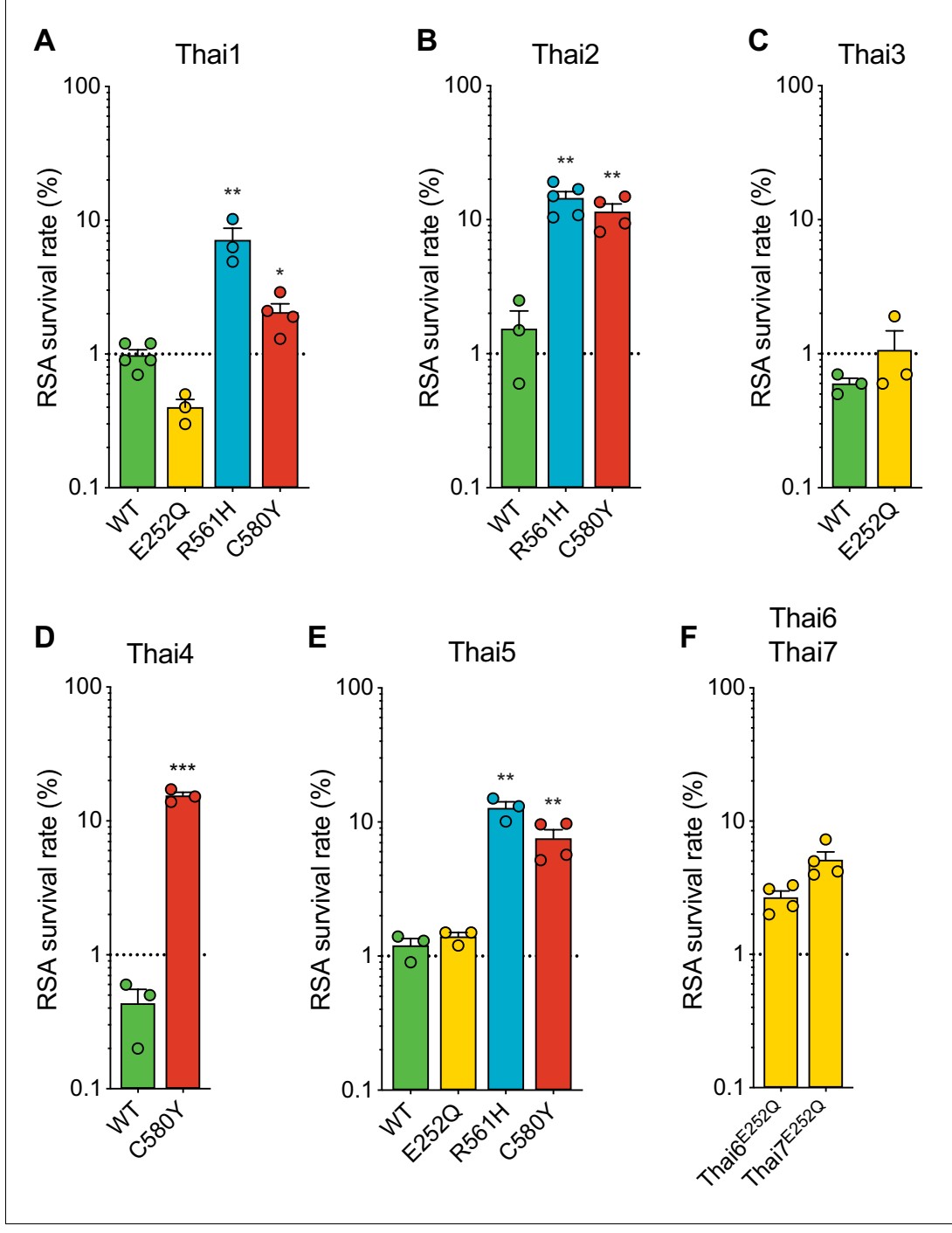

**Figure 6.** Thai isolates expressing mutant K13 display variable RSA survival rates. RSA survival rates for (A–E) *K13* edited Thai isolates and (F) K13 E252Q unedited Thai lines, shown as means ± SEM (detailed in *Figure 6— source data 1*). Results were obtained from three to seven independent experiments, each performed in duplicate. p Values were determined by unpaired *t* tests and were calculated for mutant lines relative to the isogenic line expressing wild-type *K13*. * p<0.05; ** p<0.01; *** p<0.001.

The online version of this article includes the following source data for figure 6:

**Source data 1.** Ring-stage survival (RSA) assay data for *K13* edited Thai parasites and controls.

performed in the Cambodian K13 C580Y strains RF7$^{C580Y}$ and Cam3.II$^{C580Y}$. Isogenic RF7$^{C580Y}$ parasites expressing either the mutant or wild-type fd residue at position 193 showed no difference in RSA survival rates, either at 700 nM (averaging ~27%), or across a range of DHA concentrations down to 1.4 nM (*Figure 7A,C*; *Figure 7—figure supplement 1*; *Figure 7—source data 1*). Editing fd D193Y into the recombinant CamWT$^{C580Y}$ line that expresses K13 C580Y (*Straimer et al., 2015*) also had no impact on RSA survival (mean RSA survival rate of 11% vs 12.5%). Likewise, Cam3.II$^{C580Y}$ parasites maintained the same rates of in vitro RSA survival (means ~19–22%) irrespective of their *mdr2* allele. Silent shield mutations had no impact on RSA survival rates for either *fd* or *mdr2*. eGFP-based fitness assays initiated at different starting ratios of the Dd2 eGFP$^{+}$ reporter line and *fd* or *mdr2* edited RF7$^{C580Y}$ or Cam3.II$^{C580Y}$ lines revealed no changes in growth rates for mutant lines compared with their wild-type controls (*Figure 7B,D*; *Figure 7—figure supplement 1*; *Figure 7—source data 2* and *3*). These data suggest that the fd D193Y and mdr2 T484I mutations may be markers of ART-resistant founder populations, but themselves do not contribute directly to ART resistance or augment parasite fitness.

## Discussion

Mutant K13-mediated ART resistance has substantially compromised the efficacy of antimalarial treatments across SE Asia, and the relatively high prevalence of the R561H variant that has recently been associated with delayed clearance in Rwanda highlights the risk of ART resistance emerging and spreading in sub-Saharan Africa (*Uwimana et al., 2020*; *Bergmann et al., 2021*; *Uwimana et al., 2021*). Using gene editing and phenotypic analyses, we provide definitive evidence that the K13 R561H, M579I and C580Y mutations can confer in vitro ART resistance in several African strains. In vitro resistance, as defined using the RSA, was comparable between gene-edited K13 R561H 3D7 parasites (originating from or near Rwanda) and C580Y Dd2 and Cam3.II parasites (from SE Asia). Further investigations into edited African 3D7 parasites showed that these mutations also resulted in an ~30% decrease in K13 protein levels, consistent with earlier studies into the mechanistic basis of mutant K13-mediated ART resistance (*Birnbaum et al., 2017*; *Siddiqui et al., 2017*; *Yang et al., 2019*; *Gnädig et al., 2020*; *Mok et al., 2021*). We also observed that K13 mutant African strains differed widely in their RSA survival rates. As an example, when introduced into the Tanzanian F32 strain, the C580Y mutation yielded a 0.3% RSA survival rate, contrasting with 11.8% survival in the Ugandan UG815 strain. These data suggest that F32 parasites lack additional genetic determinants that are required for mutant K13 to confer ART resistance. Collectively, our results provide evidence that certain African strains present no major biological obstacle to becoming ART resistant in vitro upon acquiring K13 mutations. Further gene editing experiments are merited to extend these studies to additional African strains, and to incorporate other variants such as C469Y and A675V that are increasing in prevalence in Uganda (*Asua et al., 2021*).

Our mixed culture competition assays with African parasites revealed substantial fitness costs with the K13 C580Y mutation in three of the four strains tested (UG659 was the exception). The largest growth defect was observed with the edited UG815 C580Y line, which also yielded the highest level of ART resistance in vitro. These data suggest that K13 C580Y may not easily take hold in Africa where, unlike in SE Asia, infections are often highly polyclonal, generating intra-host competition that impacts a strain's ability to succeed at the population level. In addition, individuals in highly-endemic African settings generally have high levels of acquired immunity, potentially minimizing infection by relatively unfit parasites, and often have asymptomatic infections that go untreated and are thus less subject to selective drug pressure compared with individuals in SE Asia (*Eastman and Fidock, 2009*). This situation recalls the history of chloroquine use in Africa, where fitness costs caused by mutations in the primary resistance determinant PfCRT resulted in the rapid resurgence of wild-type parasites following the implementation of other first-line antimalarial therapies (*Kublin et al., 2003*; *Laufer et al., 2006*; *Ord et al., 2007*; *Frosch et al., 2014*).

An even greater fitness cost was observed with the M579I mutation, earlier detected in an infection acquired in Equatorial Guinea with evidence of in vivo ART resistance (*Lu et al., 2017*), but which was notably absent in all 3257 African samples reported herein. In contrast, we observed no evident fitness cost in 3D7 parasites expressing the R561H variant, which might help contribute to its increasing prevalence in Rwanda. While our Rwandan samples from 2012 to 2015 observed this mutation at 2% prevalence, samples collected by others in 2018 and 2019 identified this mutation at

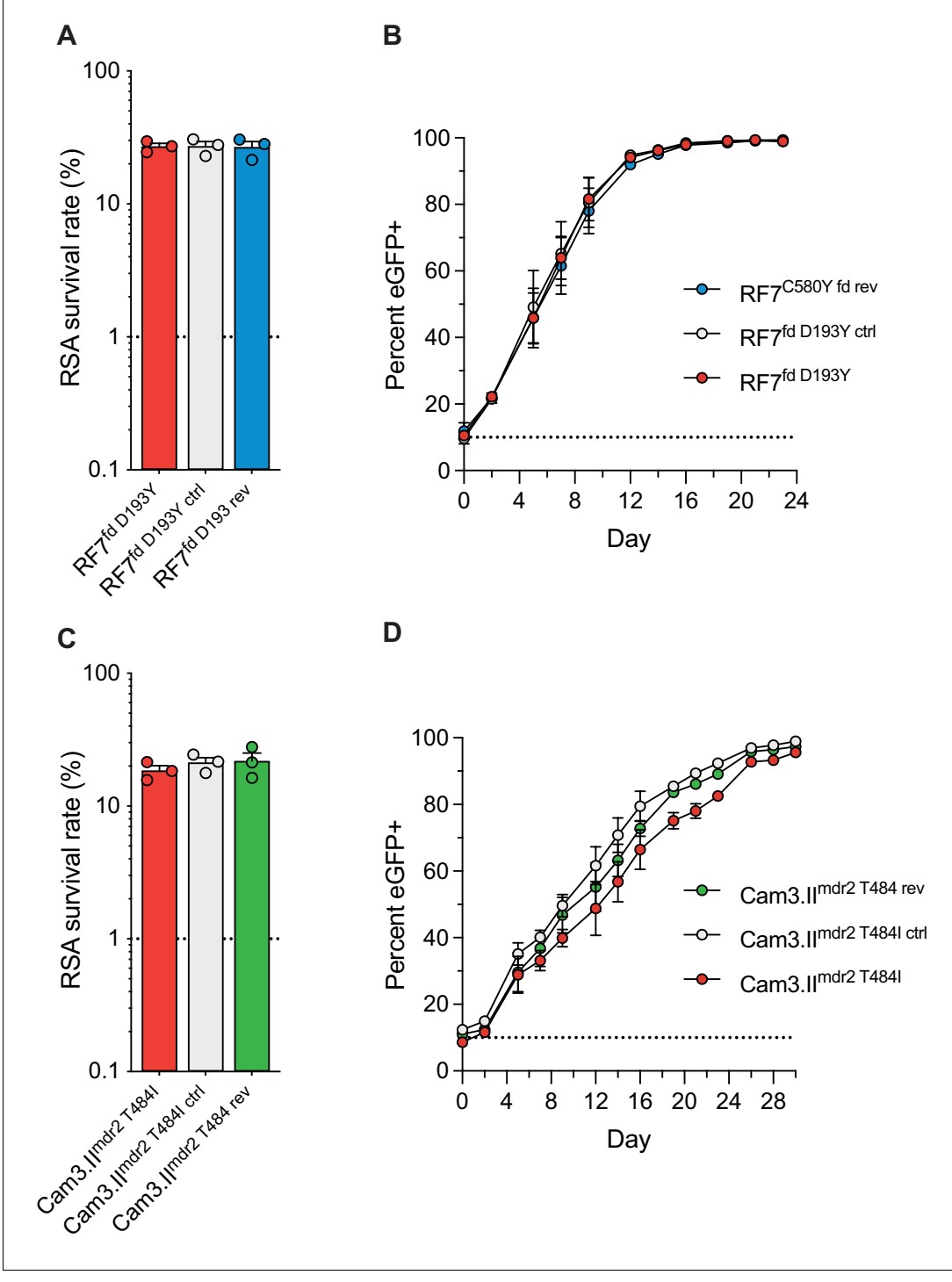

**Figure 7.** Ferredoxin (fd) and multidrug resistance protein 2 (mdr2) mutations do not impact RSA survival or in vitro growth rates in K13 C580Y parasites. RSA survival rates for (**A**) RF7[C580Y] parasite lines expressing the fd variant D193Y (parent), this variant plus silent shield mutations (edited control), or fd D193 (edited revertant), and (**C**) Cam3.II[C580Y] parasite lines expressing the mdr2 variant T484I (parent), this variant plus silent shield mutations (edited control), or mdr2 T484 (edited revertant). Parental lines are described in **Table 1** and **Supplementary file 4**. Mean ± SEM survival rates were generated from three independent experiments, each performed in duplicate. (**B, D**) In vitro eGFP-based fitness assays performed with (**B**) fd and (**D**) mdr2 RF7[C580Y] or Cam3.II[C580Y] edited lines, respectively. Competitive growth assays were seeded with individual lines plus the Dd2eGFP[+] reporter line at starting ratios of 10:1. Results show percentages of eGFP[+] parasites over time. Co-cultures were maintained over a

*Figure 7 continued on next page*

*Figure 7 continued*

period of 24 days (*fd* edited lines) or 30 days (*mdr2* edited lines), and percentages of eGFP⁺ parasites were determined by flow cytometry. Results were obtained from two to three independent experiments, each performed in triplicate, and are shown as means ± SEM. All values are provided in *Figure 7—source data 1*, *2*, *3*. The online version of this article includes the following source data and figure supplement(s) for figure 7:

**Source data 1.** Ring-stage survival (RSA) assay data for *fd* and *mdr2* edited parasites and controls.
**Source data 2.** Fitness assay data (percent eGFP+ parasites) for RF7 *fd* edited parasites and parental control.
**Source data 3.** Fitness assay data (percent eGFP+ parasites) for Cam3.II *mdr2* edited parasites and parental control.
**Figure supplement 1.** Ferredoxin (fd) and multidrug resistance protein 2 (mdr2) mutations do not impact RSA survival or in vitro growth in K13 C580Y parasites.

12–13% prevalence (*Bergmann et al., 2021*; *Uwimana et al., 2021*). One of these reports included evidence associating R561H with delayed parasite clearance in patients treated with the ACT artemether-lumefantrine (*Uwimana et al., 2021*). These recent data heighten the concern that mutant K13 might be taking hold in certain areas in Africa where it may begin to compromise ACT efficacy.

In Cambodia, our spatio-temporal analysis of K13 sequence diversity highlights the initial emergence of C580Y in the western provinces, and its progressive replacement of other variants in the country. Interestingly, this mutation was already at high prevalence in western Cambodia several years before the first published reports of delayed parasite clearance in ART-treated patients (*Noedl et al., 2008*; *Dondorp et al., 2009*; *Ariey et al., 2014*). The success of this mutation in Cambodia, and elsewhere in the eastern Greater Mekong subregion (*Imwong et al., 2020*), cannot be explained by resistance alone, as we previously reported that the less common R539T and I543T variants conferred greater ART resistance in vitro (*Straimer et al., 2015*). Similarly, we now report that the R561H and P553L mutations yield ART resistance at levels comparable to C580Y in Dd2 parasites. In contrast, low-level resistance was observed with F446I, which has nonetheless spread across Myanmar (*Imwong et al., 2020*). In a separate recent gene editing study, F446I yielded no significant in vitro resistance in 3D7 parasites and was fitness neutral (*Siddiqui et al., 2020*), consistent with our findings for this mutation in edited Dd2 parasites.

Our studies into the impact of K13 mutations on in vitro growth in Asian Dd2 parasites provide evidence that the C580Y mutation generally exerts less of a fitness cost relative to other K13 variants, as measured in *K13* edited parasites co-cultured with an eGFP⁺ reporter line. A notable exception was P553L, which compared with C580Y was similarly fitness neutral and showed similar RSA values. P553L has nonetheless proven far less successful in its regional dissemination compared with C580Y (*Ménard et al., 2016*). These data suggest that additional factors have contributed to the success of C580Y in sweeping across SE Asia. These might include specific genetic backgrounds that have favored the dissemination of C580Y parasites, possibly resulting in enhanced transmission potential (*Witmer et al., 2020*), or ACT use that favored the selection of partner drug resistance in these parasite backgrounds (*van der Pluijm et al., 2019*). In terms of growth rates in our isogenic Dd2 lines, the most detrimental impacts were observed with E252Q and R561H, which earlier predominated near the Thailand-Myanmar border region, but were later overtaken by C580Y (*Phyo et al., 2016*). In our study, C580Y produced an optimal combination of no measurable fitness cost and relatively high RSA survival rates in Dd2 parasites. In a prior independent study, however, R561H showed slightly improved fitness relative to C580Y in paired isogenic parasites from Thailand (generated in the NHP4302 strain), providing further evidence that both fitness and resistance are strain-dependent (*Nair et al., 2018*).

Further research is required to define secondary genetic determinants that could augment mutant K13-mediated ART resistance, and to explore other potential mediators of resistance. Proposed candidates have included *fd*, *mdr2*, *ap-2μ*, *ubp1*, and *pfcoronin*, which have earlier been associated with *P. falciparum* ART susceptibility (*Demas et al., 2018*; *Henrici et al., 2019*; *Sutherland et al., 2021*). Our data argue against a direct role for mutations in *fd* and *mdr2* in the strains tested herein. We also observed no evident association between the genotypes of *pfcrt*, *mdr1*, *arps10*, *ap-2μ*, or *ubp1* and the degree to which mutant K13 conferred ART resistance in vitro in our set of African or Asian strains (*Supplementary file 4*). Mutations associated with enhanced DNA repair mechanisms have also been observed in ART-resistant SE Asian parasites, supporting the idea that mutant K13 parasites may have an improved ability to repair ART-mediated DNA damage (*Xiong et al., 2020*).

Additional studies are merited to investigate whether these DNA repair mutations may provide a favorable background for the development of ART resistance.

At the population level, we note that *P. falciparum* genomic structures in Africa tend to be far more diverse than in the epicenter of resistance in Cambodia, where parasite strains are highly substructured into a few lineages that can readily maintain complex genetic traits (*Amato et al., 2018*). A requirement to transmit mutant K13 and additional determinants of resistance in African malaria-endemic settings, where genetic outcrossing is the norm, would predict that ART resistance will spread more gradually on this continent than in SE Asia. It is nonetheless possible that secondary determinants will allow some African strains to offset fitness costs associated with mutant K13, or otherwise augment K13-mediated ART resistance. Identifying such determinants could be possible using genome-wide association studies or genetic crosses between ART-resistant and sensitive African parasites in the human liver-chimeric mouse model of *P. falciparum* infection (*Vaughan et al., 2015*; *Amambua-Ngwa et al., 2019*). Reduced transmission rates in areas of Africa where malaria is declining, leading to lower levels of immunity, may also benefit the emergence and dissemination of mutant K13 (*Conrad and Rosenthal, 2019*).

Another impediment to the dissemination of ART resistance in Africa is the continued potent activity of lumefantrine, the partner drug in the first line treatment artemether-lumefantrine (*Conrad and Rosenthal, 2019*). This situation contrasts with SE Asia where ART-resistant parasites have also developed high-level resistance to the partner drug PPQ, with widespread treatment failures enabling the dissemination of multidrug-resistant strains (*Conrad and Rosenthal, 2019*; *van der Pluijm et al., 2019*). While the genotyping data presented herein and other recent molecular surveillance studies reveal a low prevalence of mutant K13 in Africa (*Kayiba et al., 2021*; *Schmedes et al., 2021*), the emergence and spread of the R561H variant in Rwanda is cause for significant concern. These data call for continuous continent-wide monitoring of the emergence and spread of mutant K13 in Africa, and for studies into whether its emergence in Rwanda is a harbinger of subsequent partner drug resistance and ACT treatment failure.

# Materials and methods

## Key resources table

| Reagent type (species) or resource | Designation | Source or reference | Identifiers | Additional information |
|---|---|---|---|---|
| Gene (*Plasmodium falciparum* 3D7 strain) | *Kelch13 (K13)* | PlasmoDB | PF3D7_1343700 | |
| Gene (*Plasmodium falciparum* 3D7 strain) | *Ferredoxin (fd)* | PlasmoDB | PF3D7_1318100 | |
| Gene (*Plasmodium falciparum* 3D7 strain) | *Multidrug resistance protein 2 (mdr2)* | PlasmoDB | PF3D7_1447900 | |
| Strain, strain background (*Plasmodium falciparum*) | 3D7 clone A10 (3D7[WT]) | D. Goldberg, Washington University School of Medicine, St. Louis, MO, USA | | see *Table 1* and *Supplementary file 4* for additional details on all *P. falciparum* strains employed herein |
| Strain, strain background (*Plasmodium falciparum*) | F32-TEM (F32[WT]) | F. Benoit-Vical, Université de Toulouse, Toulouse, France *Ariey et al., 2014* | | |

*Continued on next page*

*Continued*

| Reagent type (species) or resource | Designation | Source or reference | Identifiers | Additional information |
|---|---|---|---|---|
| Strain, strain background (*Plasmodium falciparum*) | UG659 (UG659$^{WT}$) | P. Rosenthal, University of California, San Francisco, CA, USA | | |
| Strain, strain background (*Plasmodium falciparum*) | UG815 (UG815$^{WT}$) | P. Rosenthal, University of California, San Francisco, CA, USA | | |
| Strain, strain background (*Plasmodium falciparum*) | Dd2 (Dd2$^{WT}$) | The Malaria Research and Reference Reagent Resource Center (MR4), BEI Resources | MRA-156 | |
| Strain, strain background (*Plasmodium falciparum*) | Cam3.II (Cam3.II$^{R539T}$) | R. Fairhurst, NIAID, NIH, Bethesda, MD, USA *Straimer et al., 2015* | PH0306-C | |
| Strain, strain background (*Plasmodium falciparum*) | CamWT | R. Fairhurst, NIAID, NIH, Bethesda, MD, USA *Straimer et al., 2015* | PH0164-C | |
| Strain, strain background (*Plasmodium falciparum*) | RF7 (RF7$^{C580Y}$) | R. Fairhurst, NIAID, NIH, Bethesda, MD, USA *Ross et al., 2018* | PH1008-C | |
| Strain, strain background (*Plasmodium falciparum*) | Thai1$^{WT}$ | T. Anderson, Texas Biomedical Research Institute, San Antonio, TX, USA | TA32A2A4 | |
| Strain, strain background (*Plasmodium falciparum*) | Thai2$^{WT}$ | T. Anderson, Texas Biomedical Research Institute, San Antonio, TX, USA | TA50A2B2 | |
| Strain, strain background (*Plasmodium falciparum*) | Thai3$^{WT}$ | T. Anderson, Texas Biomedical Research Institute, San Antonio, TX, USA | TA85R1 | |
| Strain, strain background (*Plasmodium falciparum*) | Thai4$^{WT}$ | T. Anderson, Texas Biomedical Research Institute, San Antonio, TX, USA | TA86A3 | |
| Strain, strain background (*Plasmodium falciparum*) | Thai5$^{WT}$ | T. Anderson, Texas Biomedical Research Institute, San Antonio, TX, USA | NHP-01334-6B | |
| Strain, strain background (*Plasmodium falciparum*) | Thai6$^{E252Q}$ | T. Anderson, Texas Biomedical Research Institute, San Antonio, TX, USA | NHP4076 | |

*Continued on next page*

*Continued*

| Reagent type (species) or resource | Designation | Source or reference | Identifiers | Additional information |
|---|---|---|---|---|
| Strain, strain background (*Plasmodium falciparum*) | Thai7$^{E252Q}$ | T. Anderson, Texas Biomedical Research Institute, San Antonio, TX, USA | NHP4673 | |
| Strain, strain background (*Escherichia coli*) | HST08 | Takara | Cat. #636766 | Stellar Competent Cells |
| Genetic reagent (*Plasmodium falciparum*) | Transgenic parasite lines | This study and *Straimer et al., 2015* | See *Supplementary file 5* | Available from D. Fidock upon request |
| Commercial assay or kit | In-Fusion HD Cloning Plus kit | Takara | Cat. #638909 | |
| Commercial assay or kit | QuantiFast Multiplex PCR Kit | Qiagen | Cat. #204654 | |
| Sequence-based reagents | Oligonucleotides | This study | See *Supplementary file 7* | |
| Recombinant DNA reagents | Plasmids | This study | See *Supplementary file 8* | Available from D. Fidock upon request |
| Sequence-based reagents | qPCR primers and probes | This study | See *Supplementary file 9* | |
| Antibody | Anti-K13 (*P. falciparum*) (Mouse monoclonal) | I. Trakht, Columbia University Medical Center, New York, NY, USA *Gnädig et al., 2020* | | Antibody clone E9 WB (1:1000) |
| Antibody | Anti-ERD2 (*P. falciparum*) (Rabbit polyclonal) | MR4, BEI Resources | MRA-1 | WB (1:1000) |
| Antibody | StarBright Blue 700 goat anti-mouse | Bio-Rad | 12004158 | WB (1:200) |
| Antibody | StarBright Blue 520 goat anti-rabbit | Bio-Rad | 12005869 | WB (1:1000) |
| Other | 4–20% Criterion TGX Precast Protein Gel | Bio-Rad | 5671093 | Used with recommended buffers, also purchased from Bio-Rad |
| Chemical compound, drug | Carbenicillin disodium salt | Sigma | C1389 | |
| Chemical compound, drug | WR99210 | Jacobus Pharmaceuticals | | |
| Chemical compound, drug | Dihydroartemisinin (DHA) | Sigma | D7439 | |
| Software, algorithm | GraphPad Prism Version 9 | GraphPad Software, San Diego, CA, USA | | graphpad.com |
| Software, algorithm | ImageJ software | NIH, Bethesda, MD, USA | | imagej.nih.gov |

## Sample collection and *K13* genotyping

Samples were obtained as blood-spot filter papers from patients seeking treatment at sites involved in national surveys of antimalarial drug resistance, or patients enrolled in therapeutic efficacy studies, or asymptomatic participants enrolled in surveillance programs. Collection details for African and Cambodian samples are provided in *Figure 1—source data 1* and *Figure 4—source data 1*, respectively. Samples were processed at the Pasteur Institute in Paris or the Pasteur Institute in Cambodia, as detailed in *Supplementary file 1*. These investigators vouch for the accuracy and completeness

of the molecular data. DNA was extracted from dried blood spots using QIAmp Mini kits, as described previously (*Ménard et al., 2016*). A nested PCR was performed on each sample to amplify the *K13* propeller domain sequence, corresponding to codons 440–680. PCR products were sequenced using internal primers and electropherograms were analyzed on both strands using the Pf3D7_1343700 3D7 sequence as the wild-type reference. Quality controls included adding six blinded quality-control samples to each 96-well sequencing plate prepared from samples from each in-country partner, and independently retesting randomly selected blood samples. Isolates with mixed alleles were considered to be mutant for the purposes of estimating mutation frequencies.

## *P. falciparum* parasite in vitro culture

*P. falciparum* asexual blood-stage parasites were cultured in human erythrocytes at 3% hematocrit in RPMI-1640 medium supplemented with 2 mM L-glutamine, 50 mg/L hypoxanthine, 25 mM HEPES, 0.21% NaHCO3, 10 mg/L gentamycin, and 0.5% w/v Albumax II (Invitrogen). Parasites were maintained at 37°C in 5% $O_2$, 5% $CO_2$, and 90% $N_2$. The geographic origins and years of culture adaptation for lines employed herein are described in *Supplementary file 4*. Parasite lines were authenticated by genotyping resistance genes and were screened by PCR for Mycoplasma every 3–6 months.

## Whole-genome sequencing of parental lines

To define the genome sequences of *P. falciparum* lines used for transfection, we lysed parasites in 0.05% saponin, washed them with 1×PBS, and purified genomic DNA (gDNA) using the QIAamp DNA Blood Midi Kit (Qiagen). DNA concentrations were quantified by NanoDrop (Thermo Scientific) and Qubit (Invitrogen) prior to sequencing. 200 ng of gDNA was used to prepare sequencing libraries using the Illumina Nextera DNA Flex library prep kit with dual indices. Samples were multiplexed and sequenced on an Illumina MiSeq to obtain 300 bp paired-end reads at an average of 50× depth of coverage. Sequence reads were aligned to the *P. falciparum* 3D7 reference genome (PlasmoDB version 36) using Burrow-Wheeler Alignment. PCR duplicates and unmapped reads were filtered out using Samtools and Picard. Reads were realigned around indels using GATK RealignerTargetCreator, and base quality scores were recalibrated using GATK BaseRecalibrator. GATK HaplotypeCaller (version 3.8) was used to identify all single nucleotide polymorphisms (SNPs). SNPs were filtered based on quality scores (variant quality as function of depth QD >1.5, mapping quality >40, min base quality score >18) and read depth (>5) to obtain high-quality SNPs, which were annotated using snpEFF. Integrated Genome Viewer was used to visually verify the presence of SNPs. BIC-Seq was used to check for copy number variations using the Bayesian statistical model (*Xi et al., 2011*). Copy number variations in highly polymorphic surface antigens and multi-gene families were removed as these are prone to stochastic changes during in vitro culture.

Whole-genome sequencing data were used to determine the genotypes of the antimalarial drug resistance loci *pfcrt*, *mdr1*, *dhfr*, and *dhps* (*Haldar et al., 2018*). We also genotyped *fd*, *arps10*, *mdr2*, *ubp1*, and *ap-2μ*, which were previously associated with ART resistance (*Henriques et al., 2014*; *Miotto et al., 2015*; *Cerqueira et al., 2017*; *Adams et al., 2018*). These results are described in *Supplementary file 4*.

## Cloning of *K13*, *fd*, and *mdr2* plasmids

Zinc-finger nuclease-meditated editing of select mutations in the *K13* locus was performed as previously described (*Straimer et al., 2015*). CRISPR/Cas9 editing of *K13* mutations was achieved using the pDC2-cam-coSpCas9-U6-gRNA-h*dhfr* all-in-one plasmid that contains a *P. falciparum* codon-optimized Cas9 sequence, a human dihydrofolate reductase (h*dhfr*) gene expression cassette (conferring resistance to WR99210) and restriction enzyme insertion sites for the guide RNA (gRNA) and donor template (*White et al., 2019*). A *K13* propeller domain-specific gRNA was introduced into this vector at the BbsI restriction sites using the primer pair p1+p2 (*Supplementary file 7*) using T4 DNA ligase (New England BioLabs). Oligos were phosphorylated and annealed prior to cloning. A donor template consisting of a 1.5 kb region of the *K13* coding region including the entire propeller domain was amplified using the primer pair p3+p4 and cloned into the pGEM T-easy vector system (Promega). This donor sequence was subjected to site-directed mutagenesis in the pGEM vector to introduce silent shield mutations at the Cas9 cleavage site using the primer pair p5+p6, and to

introduce allele-specific mutations using primer pairs (p7 to p20). *K13* donor sequences were amplified from the pGEM vector using the primer pair p21+p22 and sub-cloned into the pDC2-cam-coSpCas9-U6-gRNA-h*dhfr* plasmid at the EcoRI and AatII restriction sites by In-Fusion Cloning (Takara). Final plasmids (see *Supplementary file 8*) were sequenced using primers p23 to p25. A schematic showing the method of *K13* plasmid construction can be found in *Supplementary file 2*. Both our customized zinc-finger nuclease and CRISPR/Cas9 approaches generated the desired amino acid substitutions without the genomic integration of any plasmid sequences or any additional amino acid changes in the K13 locus, and thus provide fully comparable data.

CRISPR/Cas9 editing of *fd* and *mdr2* was performed using a separate all-in-one plasmid, pDC2-cam-Cas9-U6-gRNA-h*dhfr*, generated prior to the development of the codon-optimized version used above for *K13* (*Lim et al., 2016*). Cloning was performed as for *K13*, except for gRNA cloning that was performed using In-Fusion cloning (Takara) rather than T4 ligase. Cloning of gRNAs was performed using primer pair p29+p30 for *fd* and p42+p43 for *mdr2*. Donor templates were amplified and cloned into the final vector using the primer pairs p31+p32 for *fd* and p44+p45 for *mdr2*. Site-directed mutagenesis was performed using the allele-specific primer pairs p33+p34 or p35+p36 for *fd*, and p46+p47 or p48+p49 for *mdr2*. All final plasmids (for both *fd* and *mdr2*, see *Supplementary file 8*) were sequenced using the primer pair p37+p38 (*Supplementary file 7*). Schematic representations of final plasmids are shown in *Supplementary file 6*.

## Generation of *K13*, *fd,* and *mdr2* gene-edited parasite lines

Gene-edited lines were generated by electroporating ring-stage parasites at 5–10% parasitemia with 50 µg of purified circular plasmid DNA resuspended in Cytomix. Transfected parasites were selected by culturing in the presence of WR99210 (Jacobus Pharmaceuticals) for six days post electroporation. Parental lines harboring 2–3 mutations in the *P. falciparum dhfr* gene were exposed to 2.5 nM WR99210, while parasites harboring four *dhfr* mutations were selected under 10 nM WR99210 (see *Supplementary file 4* for *dhfr* genotypes of transfected lines). Parasite cultures were monitored for recrudescence by microscopy for up six weeks post electroporation. To test for successful editing, the *K13* locus was amplified directly from parasitized whole blood using the primer pair p26+p27 (*Supplementary file 7*) and the MyTaq Blood-PCR Kit (Bioline). Primer pairs p39+p40 and p50+p51 were used to amplify *fd* and *mdr2*, respectively. PCR products were submitted for Sanger sequencing using the PCR primers as well as primer p28 in the case of *K13*, p41 (*fd*) or p52 (*mdr2*). Bulk-transfected cultures showing evidence of editing by Sanger sequencing were cloned by limiting dilution. All gene-edited transgenic lines generated herein are described in *Supplementary file 5*.

## Parasite synchronization, ring-stage survival assays (RSAs), and flow cytometry

Synchronized parasite cultures were obtained by exposing predominantly ring-stage cultures to 5% D-Sorbitol (Sigma) for 15 min at 37°C to remove mature parasites. After 36 hr of subsequent culture, multinucleated schizonts were purified over a density gradient consisting of 75% Percoll (Sigma). Purified schizonts were incubated with fresh RBCs for 3 hr, and early rings (0–3 hr post invasion; hpi) were treated with 5% D-Sorbitol to remove remaining schizonts.

In vitro RSAs were conducted as previously described, with minor adaptations (*Straimer et al., 2015*). Briefly, tightly synchronized 0–3 hpi rings were exposed to a pharmacologically-relevant dose of 700 nM DHA or 0.1% dimethyl sulfoxide (DMSO; vehicle control) for 6 hr at 1% parasitemia and 2% hematocrit, washed three times with RPMI medium to remove drug, transferred to fresh 96-well plates, and cultured for an additional 66 hr in drug-free medium. Removal of media and resuspension of parasite cultures was performed on a Freedom Evo 100 liquid-handling instrument (Tecan). Parasitemias were measured at 72 hr by flow cytometry (see below) with at least 100,000 events captured per sample. Parasite survival was expressed as the percentage value of the parasitemia in DHA-treated samples divided by the parasitemia in DMSO-treated samples processed in parallel. We considered any RSA mean survival rates <2% to be ART sensitive.

Flow cytometry was performed on an BD Accuri C6 Plus cytometer with a HyperCyt plate sampling attachment (IntelliCyt), or on an iQue Screener Plus cytometer (Sartorius). Cells were stained with 1×SYBR Green (ThermoFisher) and 100 nM MitoTracker DeepRed (ThermoFisher) for 30 min

and diluted in 1×PBS prior to sampling. Percent parasitemia was determined as the percentage of MitoTracker⁻positive and SYBR Green-positive cells.

## Western blot analysis of K13 expression levels in edited lines

Western blots were performed with lysates from tightly synchronized rings harvested 0–6 hr post invasion. Parasite cultures were washed twice in ice-cold 1× phosphate-buffered saline (PBS), and parasites were isolated by treatment with 0.05% saponin in PBS. Released parasites were lysed in 4% SDS, 0.5% Triton X-100 and 0.5% PBS supplemented with 1× protease inhibitors (Halt Protease Inhibitors Cocktail, ThermoFisher). Samples were centrifuged at 14,000 rpm for 10 min to pellet cellular debris. Supernatants were collected and protein concentrations were determined using the DC protein assay kit (Bio-Rad). Laemmli Sample Buffer (Bio-Rad) was added to lysates and samples were denatured at 90°C for 10 min. Proteins were electrophoresed on precast 4–20% Tris-Glycine gels (Bio-Rad) and transferred onto nitrocellulose membranes. Western blots were probed with a 1:1000 dilution of primary antibodies to K13 (*Gnädig et al., 2020*) or the loading control ERD2 (BEI Resources), followed by a 1:200 dilution of fluorescent StarBright secondary antibodies (Bio-Rad). Western blots were imaged on a ChemiDoc system (Bio-Rad) and band intensities quantified using ImageJ.

## TaqMan allelic discrimination real-time (quantitative) PCR-based fitness assays

Fitness assays with African *K13* edited parasite lines were performed by co-culturing isogenic wild-type unedited and mutant edited parasites in 1:1 ratios. Assays were initiated with tightly synchronized trophozoites. Final culture volumes were 3 mL. Cultures were maintained in 12-well plates and monitored every four days over a period of 36 days (18 generations) by harvesting at each time point a fraction of each co-culture for saponin lysis. gDNA was then extracted using the QIAamp DNA Blood Mini Kit (Qiagen). The percentage of the wild-type or mutant allele in each sample was determined in TaqMan allelic discrimination real-time PCR assays. TaqMan primers (forward and reverse) and TaqMan fluorescence-labeled minor groove binder probes (FAM or HEX, Eurofins) are described in *Supplementary file 9*. Probes were designed to specifically detect the K13 R561H, M579I, or C580Y propeller mutations. The efficiency and sensitivity of the TaqMan primers was assessed using standard curves comprising 10-fold serially diluted templates ranging from 10 ng to 0.001 ng. Robustness was demonstrated by high efficiency (88–95%) and $R^2$ values (0.98–1.00). The quantitative accuracy in genotype calling was assessed by performing multiplex qPCR assays using mixtures of wild-type and mutant plasmids in fixed ratios (0:100, 20:80, 40:60, 50:50, 60:40, 80:20, 100:0). Triplicate data points clustered tightly, indicating high reproducibility in the data across the fitted curve ($R^2$ = 0.89–0.91).

Purified gDNA from fitness co-cultures was subsequently amplified and labeled using the primers and probes described in *Supplementary file 9*. qPCR reactions for each sample were run in triplicate. 20 μL reactions consisted of 1×QuantiFAST reaction mix containing ROX reference dye (Qiagen), 0.66 μM forward and reverse primers, 0.16 μM FAM-MGB and HEX-MGB TaqMan probes, and 10 ng genomic DNA. Amplification and detection of fluorescence were carried out on a QuantStudio3 qPCR machine (Applied Biosystems) using the genotyping assay mode. Cycling conditions were as follows: 30 s at 60°C; 5 min at 95°C; and 40 cycles of 30 s at 95°C and 1 min at 60°C for primer annealing and extension. Every assay was run with positive controls (wild-type or mutant plasmids at different fixed ratios). No-template negative controls (water) in triplicates were processed in parallel. Rn, the fluorescence of the FAM or HEX probe, was normalized to the fluorescence signal of the ROX reporter dye. Background-normalized fluorescence (Rn minus baseline, or ΔRn) was calculated as a function of cycle number.

To determine the wild-type or mutant allele frequency in each sample, we first confirmed the presence of the allele by only retaining values where the threshold cycle ($C_t$) of the sample was less than the no-template control by at least three cycles. Next, we subtracted the ΔRn of the samples from the background ΔRn of the no-template negative control. We subsequently normalized the fluorescence to 100% using the positive control plasmids to obtain the percentage of the wild-type and mutant alleles for each sample. The final percentage of the mutant allele was defined as the average of these two values: the normalized percentage of the mutant allele, and 100% minus the normalized percentage of the wild-type allele.

## eGFP-based fitness assays

Fitness assays with Dd2, RF7$^{C580Y}$, and Cam3.II$^{C580Y}$ *K13, fd,* or *mdr2* edited parasite lines were performed using mixed culture competition assays with an eGFP$^+$ Dd2 reporter line (*Ross et al., 2018*). This reporter line uses a *calmodulin* (*cam*) promoter sequence to express high levels of GFP and includes h*dhfr* and *blasticidin S-deaminase* expression cassettes. This line was earlier reported to have a reduced growth rate relative to parental non-recombinant Dd2, presumably at least in part because of its high levels of GFP expression (*Ross et al., 2018*; *Dhingra et al., 2019*). With our Dd2 parasites, *K13* edited lines were co-cultured in 1:1 ratios with the reporter line. This ratio was adjusted to 10:1 or 100:1 for *fd* edited RF7$^{C580Y}$ and *mdr2* edited Cam3.II$^{C580Y}$ parasites relative to the eGFP line, given the slower rate of growth with RF7 and Cam3.II parasites. Fitness assays were initiated with tightly synchronized trophozoites in 96-well plates with 200 μL culture volumes. Percentages of eGFP$^+$ parasites were monitored by flow cytometry every two days over a period of 20 days (10 generations). Flow cytometry was performed as written above, except that only 100 nM MitoTracker DeepRed staining was used to detect total parasitemias, since SYBR Green and eGFP fluoresce in the same channel.

## Fitness costs

The fitness cost associated with a line expressing a given K13 mutation was calculated relative to its isogenic wild-type counterpart using the following equation:

$$P' = P((1 - x)^n)$$

where P′ is equal to the parasitemia at the assay endpoint, P is equal to the parasitemia on day 0, n is equal to the number of generations from the assay start to finish, and x is equal to the fitness cost. This equation assumes 100% growth for the wild-type comparator line. For qPCR and GFP-based fitness assays, days 36 and 20 were set as the assay endpoints, resulting in the number of parasite generations (n) being set to 18 and 10, respectively.

# Acknowledgements

We thank Dr. Pascal Ringwald (World Health Organization) for his support and feedback. DAF gratefully acknowledges the US National Institutes of Health (R01 AI109023), the Department of Defense (W81XWH1910086) and the Bill and Melinda Gates Foundation (OPP1201387) for their financial support. BHS was funded in part by T32 AI106711 (PD: D Fidock). SM is a recipient of a Human Frontiers of Science Program Long-Term Fellowship. CHC was supported in part by the NIH (R01 AI121558; PI: Jonathan Juliano). FN is supported by the Wellcome Trust of Great Britain (Grant ID: 106698). TJCA acknowledges funding support from the NIH (R37 AI048071). PJR received funding support from the NIH (R01 AI075045). DAF and DM gratefully acknowledge the World Health Organization for their funding. We thank the following individuals for their kind help with the *K13* genotyped samples – Chad: Ali S Djiddi, Mahamat S I Diar, Kodbessé Boulotigam, Mbanga Djimadoum, Hamit M Alio, Mahamat M H Taisso, Issa A Haggar; Burkina Faso: TES 2017–2018 team and the US President's Malaria Initiative through the Improving Malaria Care Project as the funding agency for the study in Burkina Faso, Chris-Boris G Panté-Wockama; Burundi: Dismas Baza; Tanzania: Mwaka Kakolwa, Celine Mandara, Tanzania TES coordination team for the Ministry of Health; Sierra Leone: Anitta R Y Kamara, Foday Sahr, Mohamed Samai; The Gambia: Balla Kandeh, Joseph Okebe, Serign J Ceesay, Baboucarr Babou, Emily Jagne, Alsan Jobe; Congo: Brice S Pembet, Jean M Youndouka; Somalia: Jamal Ghilan Hefzullah Amran, Abdillahi Mohamed Hassan, Abdikarim Hussein Hassan and Ali Abdulrahman; Rwanda: extended TES team for the Malaria and Other Parasitic Diseases Division, Rwanda Biomedical Centre.

# Additional information

## Funding

| Funder | Grant reference number | Author |
| --- | --- | --- |
| National Institute of Allergy | R01 AI109023 | David A Fidock |

and Infectious Diseases

| | | |
|---|---|---|
| U.S. Department of Defense | W81XWH1910086 | David A Fidock |
| Bill and Melinda Gates Foundation | OPP1201387 | David A Fidock |
| Wellcome Trust | 106698 | François Nosten |
| National Institute of Allergy and Infectious Diseases | R37 AI048071 | Timothy Anderson |
| National Institute of Allergy and Infectious Diseases | T32 AI106711 | David A Fidock |
| World Health Organization | | Didier Menard David A Fidock |
| National Institute of Allergy and Infectious Diseases | R01 AI075045 | Philip J Rosenthal |

The funders had no role in study design, data collection and interpretation, or the decision to submit the work for publication.

### Author contributions

Barbara H Stokes, Conceptualization, Data curation, Formal analysis, Validation, Investigation, Visualization, Methodology, Writing - original draft, Writing - review and editing; Satish K Dhingra, Conceptualization, Formal analysis, Investigation, Methodology; Kelly Rubiano, Data curation, Investigation; Sachel Mok, Formal analysis, Investigation, Methodology; Judith Straimer, Conceptualization, Investigation; Nina F Gnädig, Ioanna Deni, Kyra A Schindler, Jade R Bath, Kurt E Ward, Josefine Striepen, Leila S Ross, Eric Legrand, Clark H Cunningham, Investigation; Tomas Yeo, Data curation, Software, Formal analysis, Methodology; Frédéric Ariey, Software, Methodology; Issa M Souleymane, Adama Gansané, Romaric Nzoumbou-Boko, Claudette Ndayikunda, Abdunoor M Kabanywanyi, Aline Uwimana, Samuel J Smith, Olimatou Kolley, Mathieu Ndounga, Marian Warsame, Resources, Investigation; Rithea Leang, Formal analysis, Investigation; François Nosten, Timothy JC Anderson, Resources; Philip J Rosenthal, Resources, Writing - review and editing; Didier Ménard, Conceptualization, Resources, Supervision, Funding acquisition, Writing - review and editing; David A Fidock, Conceptualization, Supervision, Funding acquisition, Visualization, Writing - original draft, Project administration, Writing - review and editing

### Author ORCIDs

Barbara H Stokes (iD) https://orcid.org/0000-0001-9519-487X
Kelly Rubiano (iD) http://orcid.org/0000-0002-4149-4030
Sachel Mok (iD) http://orcid.org/0000-0002-9605-0154
Ioanna Deni (iD) http://orcid.org/0000-0002-5266-8243
Kurt E Ward (iD) http://orcid.org/0000-0002-3614-436X
Tomas Yeo (iD) http://orcid.org/0000-0003-2923-6060
Abdunoor M Kabanywanyi (iD) http://orcid.org/0000-0002-2980-0429
François Nosten (iD) http://orcid.org/0000-0002-7951-0745
David A Fidock (iD) https://orcid.org/0000-0001-6753-8938

### Ethics

Human subjects: Health care facilities were in charge of collecting anonymized *P. falciparum* positive cases. Identification of individuals cannot be established. The studies were approved by ethics committees listed in Supplementary file 1. We note that the sponsors had no role in the study design or in the collection or analysis of the data. There was no confidentiality agreement between the sponsors and the investigators.

### Decision letter and Author response

Decision letter https://doi.org/10.7554/eLife.66277.sa1
Author response https://doi.org/10.7554/eLife.66277.sa2

## Additional files

### Supplementary files

• Supplementary file 1. Sample information and approval from within-country ethics committees for *K13* genotyping data.

• Supplementary file 2. CRISPR/Cas9 strategy for editing the *K13* locus. All-in-one plasmid approach used for CRISPR/Cas9-mediated *K13* gene editing, consisting of a *K13*-specific donor template for homology-directed repair, a *K13*-specific gRNA expressed from the *U6* promoter, a Cas9 cassette with expression driven by the *calmodulin* (*cam*) promoter, and a selectable marker (human *dhfr*, conferring resistance to the antimalarial WR99210 that inhibits *P. falciparum* DHFR). The Cas9 sequence was codon-optimized for improved expression in *P. falciparum*. Donors coding for specific mutations of interest (e.g., K13 C580Y, red star) were generated by site-directed mutagenesis of the K13 wild-type donor sequence. Green bars indicate the presence of silent shield mutations that were introduced to protect the edited locus from further cleavage. The lightning bolt indicates the location of the cut site in the genomic target locus. Primers used for cloning and final plasmids are described in *Supplementary files 7* and *8*, respectively.

• Supplementary file 3. Crystal structure of K13 propeller domain showing positions of mutated residues. (A, B) Top and (C, D) side views of the crystal structure of the K13 propeller domain (PDB ID: 4YY8), highlighting residues of interest (F446I, orange; R539T, dark blue; I543T, purple; P553L, pink; R561H, dark turquoise; P574L, light turquoise; M579I medium blue; C580Y, red). Structures shown in (A) and (C) show wild-type residues while (B) and (D) show mutated residues.

• Supplementary file 4. Geographic origin and drug resistance genotypes of *P. falciparum* clinical isolates and reference lines employed in this study.

• Supplementary file 5. Transgenic *P. falciparum* lines generated in this study.

• Supplementary file 6. CRISPR/Cas9 strategy for editing the *ferredoxin* (*fd*) and *multidrug resistance protein 2* (*mdr2*) loci. All-in-one plasmid approaches used for CRISPR/Cas9-mediated editing of (A) the *ferredoxin* (*fd*) locus or (B) the *multidrug resistance protein 2* (*mdr2*) locus. Plasmids consisted of a (A) *fd* or (B) *mdr2* specific donor template for homology-directed repair, a gene-specific gRNA expressed from the *U6* promoter, a Cas9 cassette with expression driven by the *cam* promoter, and a selectable marker (human *dhfr*, conferring resistance to WR99210). Donors coding for specific mutations of interest (fd D193Y or mdr2 T484I) were generated by site-directed mutagenesis of the wild-type donor sequences. Red bars indicate the presence of silent shield mutations used to protect edited loci from further cleavage. Primers used for cloning and final plasmids are described in *Supplementary files 7* and *8*, respectively.

• Supplementary file 7. Oligonucleotides used in this study.

• Supplementary file 8. Description of gene-editing plasmids generated in this study.

• Supplementary file 9. Real-time PCR (qPCR) primers and probes used in this study.

• Transparent reporting form

### Data availability

All data generated or analysed during this study are included in the manuscript and supporting files. Source data files have been provided for Figures 1-7.

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
