## [Decision Letter]

**Acceptance summary:**

This study is of interest to the broad malaria research community and especially those who work on drug resistance. The authors provide a summary of their surveys of African and Southeast Asian *Plasmodium falciparum* parasites for the Kelch 13 gene, a marker of artemisinin resistance. The contribution of several K13 mutations to artemisinin resistance is investigated in different genetic backgrounds and confirms the lack of a barrier for the potential emergence of artemisinin resistance in African parasites. These findings are of prime importance in the context of public health perspective on managing the risk of resistance appearing in Africa.

**Decision letter after peer review:**

Thank you for submitting your article "*P. falciparum* K13 mutations present varying degrees of artemisinin resistance and reduced fitness in African parasites" for consideration by *eLife*. Your article has been reviewed by 3 peer reviewers, and the evaluation has been overseen by Dominique Soldati-Favre as the Senior and Reviewing Editor. The reviewers have opted to remain anonymous.

The reviewers have discussed their reviews with one another, and the Reviewing Editor has drafted this to help you prepare a revised submission. Beside the Essential Revisions, please consider the recommendations made by the individual reviewers and address the specific points as far as possible in the revision.

Essential revisions:

1) The authors genotyped K13 in 3,327 parasite isolates from 4 provinces of Cambodia and report changes in distribution of K13 alleles. This is a useful graphical analysis of the increasing dominance of C580Y; and tells the same story as several other recent studies of isolates from this region. It would be useful to explain the overlap (or differences) from sample sets used in other studies of this kind.

2) There is an impressive amount of work from different labs presented in this manuscript. However, a systematic analysis of the key R561H Rwandan mutations in well-validated field parasites of Asian backgrounds is lacking – making it difficult to draw firm conclusions about the relationship between fitness cost and allele persistence.

3) A couple of small experiments are requested to clarify two standing questions – one is "does ART resistance in these modified lines show lower K13 expression?" The Fidock group has the unique monoclonal antibodies to easily address this question. The second is regarding whether the background mutations help the parasites to be more fit with the K13 mutations given that it is suspected so? With the eGFP parasite in hand, this can be easily addressed. These will further support the central claims of the paper.

4) The authors state that the data provides conclusive evidence that there is no core biological obstacle to becoming Art resistance for African strains – this is indeed a very important observation however, this is based solely on RSA and in vitro culture and as the authors show themselves the genetic background of the parasite is critical. It is therefore important to qualify the statement somewhat as it is quite plausible that the genetic background of the African strains cannot overcome the fitness loss. This would raise the question on whether introduction of a SE Asian parasite genetic background poses a risk.

The authors should consider recent publications that show that DNA repair may be critical for this.

*Reviewer #1 (Recommendations for the authors):*

There are a couple of issues I think would be beneficial for the authors to consider.

The RSA is plotted on a log scale making it nearly impossible to distinguish between differences in the strains.

Efforts should be made to somehow calibrate the data from Figure 3 E and F and Figure 5 c and D so as to ensure that a reader can easily compare and contrast the relationship between RSA and fitness cost.

The extremely high RSA of R539T in Figure 5A and B is hardly discussed and interestingly it is not shown in the subsequent C and D. Can the authors explain this?

The authors state that the data provides conclusive evidence that there is no core biological obstacle to becoming Art resistance for African strains – this is indeed a very important observation however, this is based solely on RSA and in vitro culture and as the authors show themselves the genetic background of the parasite is critical. It is therefore important to qualify the statement somewhat as it is quite plausible that the genetic background of the African strains cannot overcome the fitness loss. This would raise the question on whether introduction of a SE Asian parasite genetic background poses a risk.

The authors should consider recent publications that show that DNA repair may be critical for this.

*Reviewer #2 (Recommendations for the authors):*

1. The surveillance of K13 mutations in Cambodia and Africa is a significant undertaking to monitor the emergence and spread of ART resistance. While some sampling periods overlapped with earlier survey reports (shown in Supplementary file 4), it would be nice to clarify that some samples were from earlier studies in the results part. Are the samples carrying the R561H from the same study as in Uwimana et al., 2020 or do they represent different samples?

2. Editing the M579I and C580Y mutations in the 4 African parasites demonstrated the importance of the genetic background. Also, the authors sequenced the genomes of all the parasite clones used. However, the information is only mentioned in a supplemental table. It would be nice to mention the major differences or genetic backgrounds (in terms of drug resistance genes) and speculate what differences are probably responsible for the varying degrees of RSA results.

3. The same should be explored in the SE Asian strains.

4. The authors' group and other groups have shown that the lower expression levels of K13 is responsible for the ART resistance phenotype in earlier studies.

It would be interesting to check whether the strains which show higher survival rates after introduction of K13 mutations also have a lower K13 expression.

5. In figure 5, it is shown that Dd2 eGFP parasite had considerable fitness costs, please provide a reason. Also, the data might be better presented as normalized against Dd2WT (using Dd2 eGFP as the reference).

6. The mdr2 and/or ferrodoxin mutations: given that these mutations are suspected to confer better fitness to the parasites, it makes sense to check the effect of fd D193Y and mdr2 T484I on parasite fitness.

7. In western Cambodia, WT allele percentage in 2001-2002 had already fallen to 56%, but ACT failure was detected much later. That's interesting, may be the authors can discuss this further.

8. Page 15: subtitle "Strain-dependent genetic..." needs to be revised.

9. Page 16: line 6: better to clarify as "we applied CRISPR/Cas9 editing to revert the fd D193Y and mdr2 T484I mutations to their wild type in the ... strains".

10. Given the extent of the study, the title is misleading and should be revised to reflect the large amount of data concerning the SE Asian parasites and mutations.

*Reviewer #3 (Recommendations for the authors):*

This manuscript by Stokes et al., combines a number of different studies on different parasites of different origins. It appears to be work put together from different labs conducted in some cases with different methods.

The authors have compiled genotype data from isolates from 11 different countries in Africa. The authors note the continuing dominance (99%)of wild-type K13 and the presence of the K13 R561H variant in Rwanda. The Abstract suggests that 3,299 isolates were genotyped as part of the study. However, it seems that 2,261 of these genotype datasets have been reported previously, including the 927 parasite samples from Rwanda. The authors should make this clear in the text.

It is notable that the Rwandan samples were collected from 2012 to 2015. As the authors point out more recent studies suggest that the prevalence of the K13 R561H variant remains relatively low. The authors should comment on the implications of this finding for spread of this variant.

The authors undertook further studies of three K13 mutations, R561H, C580Y and M579I. The M579I mutation was identified in a *P. falciparum*-infected individual in Equatorial Guinea who displayed delayed parasite clearance following ACT treatment. It is not clear why this variant was chosen for further study as it does not seem to have been identified in any of the 3299 samples compiled for the comparative analysis of genotypes.

Introduction of R561H, C580Y and M579I K13 mutations into F32WT (African origin) parasites yielded almost no increase in RSA survival. This is surprising given that F32 is the strain that was used in the foundational long-term ART selection studies. The original study identified the M476I K13 mutation. Gene editing studies, performed by some of the authors of the current work, were used to validate this mutation in the F32 line (survival (<0.2% in F32-TEM to 1.7% in F32-TEMM476I)). The authors state that they developed a new CRISPR/Cas9-mediated K13 editing strategy to introduce the mutations. Did the authors validate the editing strategy used in this study (and the current clone of the F32 line) by introducing the M476I K13 mutation?

The authors measured the fitness cost of the C580Y and M579I K13 mutations in parasites of African origin. These two mutations had fitness costs in all backgrounds. It is not clear why R561H was not included in this study, given that it is the only variant that appear to be persisting in an African setting. The authors conclude that the C580Y and M579I K13 mutations cause substantial fitness costs, and suggest that this may "counter-select against their dissemination in high-transmission settings". A comparison can be made with the C580Y mutation in the Dd2 (Indochina origin lab strain) background (Figure 5), but not the M579I mutation. And while there is data for R561H fitness cost in Dd2, there is no data for field strains of African and SEA backgrounds.

Another feature of the data presented in Figure 5 and 6 is that different mutations have very different fitness costs in different backgrounds, which is not readily correlated with frequency at which the alleles are found in field parasites. Moreover, in agreement with the authors' previous studies, the levels of resistance offered by different mutations in highly dependent on the genetic background. Thus, while it remains a reasonable hypothesis, the data presented here are not sufficient to support the authors' conclusion that fitness costs determine whether K13 mutations persist in the African setting.

The authors genotyped K13 in 3,327 parasite isolates from 4 provinces of Cambodia and report changes in distribution of K13 alleles. This is a useful graphical analysis of the increasing dominance of C580Y; and tells the same story as several other recent studies of isolates from this region. It would be useful to explain the overlap (or differences) from sample sets used in other studies of this kind.

Of interest, gene edited lines expressing K13 P553L in Dd2 exhibited equivalent degrees of ART resistance and a similar low fitness cost as C580Y. But P553L remains a rare mutation in the field. This suggests that the dominance of the C580Y mutation in SEA is not completely explained by resistance level conferred and fitness cost suffered, as measured by currently available assays.

In summary, while an impressive amount of work from different labs is presented in this manuscript, a systematic analysis of the key R561H Rwandan mutations in well-validated field parasites of Asian backgrounds is lacking – making it difficult to draw firm conclusions about the relationship between fitness cost and allele persistence.

---

## [Author Response]

Essential revisions:1) The authors genotyped K13 in 3,327 parasite isolates from 4 provinces of Cambodia and report changes in distribution of K13 alleles. This is a useful graphical analysis of the increasing dominance of C580Y; and tells the same story as several other recent studies of isolates from this region. It would be useful to explain the overlap (or differences) from sample sets used in other studies of this kind.

The Editors and Reviewer 3 (Comment #8) raise a valid point concerning clarification of overlaps or differences between the *K13* genotyping data presented in our manuscript and previously published datasets. With regards to the data from Cambodia shown in Figure 4, 58% of sequences were previously published. The remaining sequences (42%) are unpublished data from our coauthor Didier Ménard. These were collected at sentinel sites across four regions of Cambodia shown in Figure 4—figure supplement 1, and were sequenced at the Pasteur Institute in Phnom Penh, Cambodia (see Supplementary file 1). To more accurately reflect this information, we have added two references to Figure 4–source data 1 to indicate sources for previously published sequences (similar to Figure 1). Our revised manuscript now also clarifies the publication status of these samples as follows:

Results lines 201: “1,412 samples (42%) were obtained and sequenced during the period 2015-2017 and have not previously been published. Earlier samples were reported in (Ariey et al., 2014; Menard et al., 2016).”

Figure 4 Legend lines 961-963: “Mutations and numbers of Cambodian samples sequenced per region/year, including prior citations as appropriate, are listed in Figure 4–source data 1.”

2) There is an impressive amount of work from different labs presented in this manuscript. However, a systematic analysis of the key R561H Rwandan mutations in well-validated field parasites of Asian backgrounds is lacking – making it difficult to draw firm conclusions about the relationship between fitness cost and allele persistence.

This comment was raised by Reviewer 3, whose Comment #6 queried the absence of fitness data with R561H in field strains of African and Southeast Asian backgrounds. Our original submission did not include fitness data for the R561H variant in African parasites as this had only very recently been (the first report with data from Rwanda was from October 2020, with our original submission sent in January 2021). In response to this Reviewer’s comment, we have now completed fitness assays for R561H in the two African strains 3D7 and F32. These data, included in our revised Figure 3, provide evidence of R561H being effectively fitness neutral in 3D7 but having a fitness cost in F32 comparable to that of the C580Y or M579I mutations. Interestingly, 3D7 was recently observed to be phylogenetically closely related to Rwandan isolates (Uwimana et al., 2020, Nature Med). These data describing the impact of R561H on parasite growth in 3D7 and F32 now add to the results previously included herein on the Southeast Asian strain Dd2, as well as a prior report with an edited Thai line (Nair et al., 2018, Antimicrob Agents Chemother). In both those cases, R561H conferred a fitness cost in vitro. Our new R561H fitness data are now included and referenced in our revised manuscript as follows:

Abstract lines 34-36: “C580Y and M579I cause substantial fitness costs, which may slow their dissemination in high-transmission settings, in contrast with R561H that in African 3D7 parasites is fitness neutral.”

Results lines 169-170: “K13 C580Y and M579I mutations, but not R561H, are associated with an in vitro fitness defect across African parasites”.

Results lines 172-174: “Assays were conducted by pairing K13 wild-type lines (i.e. 3D7, F32, UG659 and UG815) with their isogenic edited R561H, M579I, or C580Y counterparts.”

Results lines 185-193: “For R561H, we observed no impact on fitness in 3D7 parasites, although in F32 this mutation exerted a fitness defect similar to M579I and C580Y (Figure 3A–D; Figure 3–source data 1). […] An exception was 3D7^R561H^ that showed moderate resistance with no apparent fitness cost (Figure 3F).”

Discussion lines 340-343: “An even greater fitness cost was obtained with the M579I mutation, earlier detected in an infection acquired in Equatorial Guinea with evidence of in vivo ART resistance (Lu et al., 2017) but which was notably absent in all 3,257 African samples reported herein. In contrast, we observed no evident fitness cost in 3D7 parasites expressing the R561H variant, which might help contribute to its increasing prevalence in Rwanda.”

Figure 3 Legend lines 945-946: This has been reworded to read “K13 mutations cause differential impacts on in vitro growth rates across gene-edited African strains.”

3) A couple of small experiments are requested to clarify two standing questions – one is "does ART resistance in these modified lines show lower K13 expression?" The Fidock group has the unique monoclonal antibodies to easily address this question. The second is regarding whether the background mutations help the parasites to be more fit with the K13 mutations given that it is suspected so? With the eGFP parasite in hand, this can be easily addressed. These will further support the central claims of the paper.

These are important suggestions, and we have now completed both sets of experiments. The Western blot data measuring K13 expression levels in our 3D7 edited lines are now shown in Figure 2—figure supplement 1 that accompanies Figure 2. These data provide evidence of reduced K13 protein levels caused by mutations that mediate ART resistance, and are consistent with the recent literature. These new data are described as follows:

Results lines 148-150: “Western blot analysis with tightly synchronized ring-stage parasites revealed a ~30% reduction in K13 protein expression levels in these three K13 mutant lines relative to the parental 3D7^WT^ (Figure 2—figure supplement 1; Figure 2—figure supplement 1–source data 1).”

Discussion lines 314-317: “Further investigations into edited African 3D7 parasites showed that these mutations also resulted in a ~30% decrease in K13 protein levels, consistent with earlier studies into the mechanistic basis of mutant K13-mediated ART resistance (Birnbaum et al., 2017; Siddiqui et al., 2017; Yang et al., 2019; Gnadig et al., 2020; Mok et al., 2021).”

Methods lines 529-540: “Western blot analysis of K13 expression levels in edited lines. Western blots were performed with lysates from tightly synchronized rings harvested 0-6 h post invasion. […] Western blots were imaged on a ChemiDoc system (Bio-Rad) and band intensities quantified using ImageJ.

Figure 2—figure supplement 1 legend lines 932-940: “African K13 mutations result in reduced K13 protein levels in 3D7 parasites. (A) Representative Western blot of parasite extracts probed with an anti-K13 monoclonal antibody (clone E9) that recognizes full-length K13 (~85 kDa) and lower molecular weight bands, presumably N-terminal degradation products, as previously reported (Gnadig et al., 2020). […] Western blots revealed reduced levels of K13 protein in the three mutant lines relative to wild-type 3D7 parasites. Results are shown as means ± SEM. WT, wild-type.”

Figure 2—figure supplement 1–source data 1 lines 942-943. “Raw figure files for K13 Western blots performed on 3D7 parasites.”

We have now also completed mixed-culture competitive growth assays with our lines that express mutant or wild-type *ferredoxin* or *mdr2*. These fitness data are shown in our revised Figure 7 and in Figure 7—figure supplement 1. Our results show no significant impact on parasite fitness with mutations in either *fd* or *mdr2* for the two strains tested herein (RF7 and Cam3.II, both K13 C580Y), supporting the interpretation that these genes could instead be simply genetic markers of the original founder populations in which mutant K13 emerged. These new data are now included in our revised manuscript as follows:

Results lines 287-288: “Mutations in the *P. falciparum* multidrug resistance protein 2 and ferredoxin genes do not modulate resistance to artemisinin or parasite fitness in vitro”

Results lines 299-304: “eGFP-based fitness assays initiated at different starting ratios of eGFP and either *fd*-edited RF7 or *mdr2*-edited Cam3.II lines revealed no change in the growth rates of the *fd* or *mdr2* mutants compared with their wild-type controls (Figure 7B, D; Figure 7—figure supplement 1; Figure 7–source data 2 and 3). These data suggest that the fd D193Y and mdr2 T484I mutations are markers of ART-resistant founder populations but themselves do not contribute directly to ART resistance or augment parasite fitness.”

Methods lines 579-586: “Fitness assays with Dd2, RF7^C580Y^ and Cam3.II parasite lines were performed using mixed culture competition assays with an eGFP-positive (eGFP^+^) Dd2 reporter line (Ross et al., 2018). […] This ratio was adjusted to 10:1 or 100:1 for *fd*-edited RF7^C580Y^ and *mdr2*-edited Cam3.II parasites relative to the eGFP line, given the slower rate of growth with RF7^C580Y^ and Cam3.II.”

Figure 7 Legend lines 1014-1026: “Figure 7. Ferredoxin (*fd)* and multidrug resistance protein 2 (*mdr2)* mutations do not impact RSA survival or in vitro growth rates in K13 C580Y parasites. […] All values are provided in Figure 7–source data 1–3.”

4) The authors state that the data provides conclusive evidence that there is no core biological obstacle to becoming Art resistance for African strains – this is indeed a very important observation however, this is based solely on RSA and in vitro culture and as the authors show themselves the genetic background of the parasite is critical. It is therefore important to qualify the statement somewhat as it is quite plausible that the genetic background of the African strains cannot overcome the fitness loss. This would raise the question on whether introduction of a SE Asian parasite genetic background poses a risk.The authors should consider recent publications that show that DNA repair may be critical for this.

We appreciate this suggestion from the Editors and Reviewer 1 and have modified our text as follows, including a reference to additional mutations of interest:

Discussion lines 321-325: “Collectively, our results provide evidence that certain African strains present no major biological obstacle to becoming ART resistant in vitro upon acquiring K13 mutations. Further gene editing experiments are merited to extend these studies to additional African strains, and to incorporate other variants such as C469Y and A675V that are increasing in prevalence in Uganda (Asua et al., 2020).”

Discussion lines 385-389: “Mutations associated with enhanced DNA repair mechanisms have also been observed in ART-resistant SE Asian parasites, supporting the idea that mutant K13 parasites may have an improved ability to repair ART-mediated DNA damage (Xiong et al., 2020). Further studies are merited to investigate whether these DNA repair mutations may provide a favorable background for the development of ART resistance.”

Reviewer #1 (Recommendations for the authors):There are a couple of issues I think would be beneficial for the authors to consider.The RSA is plotted on a log scale making it nearly impossible to distinguish between differences in the strains.

We thank the Reviewer for this suggestion regarding the log scale used to plot RSA survival values. In this case, we used log scales to plot RSA results both to reduce the amount of blank space within each RSA plot as well as to maintain consistency across previous publications from our group where RSA values have been reported (e.g. Straimer et al., 2015, Science; Gnadig et al., 2020, PLoS Pathogens). Source data for all RSA plots (i.e. means, number of repeats and statistics) are provided in the source data files for each figure, allowing for close comparisons of survival rates between lines.

Efforts should be made to somehow calibrate the data from Figure 3 E and F and Figure 5 c and D so as to ensure that a reader can easily compare and contrast the relationship between RSA and fitness cost.

Different methods were used to determine fitness costs in the gene-edited African lines and the gene-edited Dd2 lines (data presented in Figures 3 and 5, respectively). For the Dd2 lines, we made use of an existing eGFP-expressing Dd2 reporter line (Dd2^eGFP^) to perform co-culture assays with isogenic lines in order to compare growth rates across a large number of K13 variants without the need to design probes specific to each mutation. The Dd2^WT^ line was included as a control. Due to intrinsic growth defects associated with the expression of eGFP, all of the lines tested in these assays readily outcompeted the Dd2^eGFP^ reporter line. Given that there were no appreciable differences in asexual blood stage growth among the *K13* edited lines relative to the eGFP reporter line, we have now revised our manuscript to show calculated fitness costs for the K13 mutant lines relative to the control Dd2^WT^ line in revised Figures 5C and 5D. Data showing fitness costs relative to the Dd2^eGFP^ reporter line are shown in Figure 5—figure supplement 1. Our revised manuscript now reads:

Results lines 264-266: “These data provided evidence of a minimal impact with the F446I, P553L and C580Y mutations, with E252Q, R561H and P574L having greater fitness costs when compared to Dd2^WT^ (Figure 5C; Figure 5—figure supplement 1; Figure 5–source data 2).”

Figure 5 Legend lines 985-988: “Fitness costs were initially calculated relative to the Dd2^eGFP^ reporter line (Figure 5—figure supplement 1) and then normalized to the Dd2^WT^ line. Mean ± SEM values were obtained from three independent experiments, each performed in triplicate. (D) Fitness costs for K13 mutant lines, relative to the Dd2^WT^ line, were plotted against their corresponding RSA survival values.”

For Figure 3, our isogenic comparisons between K13 mutant and WT lines used the four strains 3D7, F32, UG659 and UG815. Given their different growth rates and genetic backgrounds, we decided against using the Dd2^eGFP^ reporter line as a standard, and instead we developed a separate Taqman-based allelic discrimination assay that enabled us to track allelic prevalence over time in mixed cultures containing isogenic mutant and WT lines (Figure 3). These two methods are therefore distinct, yet both allowed us to apply the same calculations to estimate the fitness cost as a percent change in prevalence of the mutant allele versus its WT control per generation (Figures 3E, 3F, 5C, 5D).

The extremely high RSA of R539T in Figure 5A and B is hardly discussed and interestingly it is not shown in the subsequent C and D. Can the authors explain this?

We included the K13 R539T mutant Dd2 and Cam3.II lines as our benchmark as our prior studies identified this as the most highly resistant K13 mutation (Straimer et al., 2015, Science). We have also published prior data on the fitness cost of R539T in two parasite strains (in that case Cam3.II and V1/S; Straimer et al., 2017, mBio). We therefore deemed it unnecessary to repeat fitness studies with this mutation herein, and focused instead on other previously untested variants. Our revised manuscript now refers to these earlier data as follows:

Results lines 243-245: “The resistant benchmark Dd2^R539T^ showed a mean RSA survival level of 20.0%, consistent with earlier reports of this mutation conferring high-grade ART resistance in vitro (Straimer et al., 2015; Straimer et al., 2017).”

The authors state that the data provides conclusive evidence that there is no core biological obstacle to becoming Art resistance for African strains – this is indeed a very important observation however, this is based solely on RSA and in vitro culture and as the authors show themselves the genetic background of the parasite is critical. It is therefore important to qualify the statement somewhat as it is quite plausible that the genetic background of the African strains cannot overcome the fitness loss. This would raise the question on whether introduction of a SE Asian parasite genetic background poses a risk.

The authors should consider recent publications that show that DNA repair may be critical for this.

Please see our response to Comment #4 from the Editors.

Reviewer #2 (Recommendations for the authors):1. The surveillance of K13 mutations in Cambodia and Africa is a significant undertaking to monitor the emergence and spread of ART resistance. While some sampling periods overlapped with earlier survey reports (shown in Supplementary file 4), it would be nice to clarify that some samples were from earlier studies in the results part. Are the samples carrying the R561H from the same study as in Uwimana et al., 2020 or do they represent different samples?

For the Cambodian samples, please see our response above to Comment #1 from the Editors. With regards to the African samples, 32% (1038/3257) of sequences were previously unpublished data. These sequences were obtained from samples collected in three countries (The Gambia, Rep. of the Congo and Burundi) that had been sent by in-country partners to our coauthor Didier Ménard for sequencing at the Pasteur Institute in Paris. All partners have consented to the publication of these data (see Supplementary file 1). The remaining sequences (including those from Rwanda that show the local emergence of the K13 R561H mutation) were previously published (see Figure 1 – source data 1) and have been compiled herein. We also note that 42 Tanzanian samples were removed from our resubmission following discussion with our partners, resulting in a final set of 3,257 African samples.

Our revised manuscript now also clarifies the origins of these sequences as follows:

Results lines 108-114: “To examine the status of K13 mutations across Africa, we analyzed *K13* beta-propeller domain sequences in 3,257 isolates from 11 malaria-endemic African countries, […] 1,038 (32%) originated from The Gambia, Republic of the Congo and Burundi and have not been previously reported, whereas the remaining samples including those from Rwanda have been published (Figure 1–Source data 1; Supplementary file 1).”

Figure 1 Legend lines 913-914: “Sample sizes and years of sample collection are indicated. Mutations and numbers of African samples sequenced per country, and prior citations as appropriate, are listed in Figure 1–source data 1.”

2. Editing the M579I and C580Y mutations in the 4 African parasites demonstrated the importance of the genetic background. Also, the authors sequenced the genomes of all the parasite clones used. However, the information is only mentioned in a supplemental table. It would be nice to mention the major differences or genetic backgrounds (in terms of drug resistance genes) and speculate what differences are probably responsible for the varying degrees of RSA results.3. The same should be explored in the SE Asian strains.

Our revised manuscript indicates the genotypes of putative or confirmed drug resistance determinants in Supplementary file 4. Aside from *K13*, none of these genes appear to correlate with the differing degrees of ART resistance in these four lines as defined using the RSA. This applies to *pfcrt* and *pfmdr1* as well as the other potential markers *arps10*, ap2-m and *ubp1*. Our study also provides evidence that *fd* and *mdr2* mutations do not contribute to ART resistance in RF7 and Cam3.II parasites, as shown in Figure 7. To identify secondary determinants that can modulate mutant K13-mediated ART resistance one could potentially conduct a *P. falciparum* genetic cross between ART-resistant and sensitive clones in the humanized mouse model (Vaughan et al., 2015, Nature Methods), however that is a multi-year project. Our revised text now expands on the genotyping of our African strains as follows:

Discussion lines 383-385: “We also observed no evident association between the genotypes of *pfcrt*, *pfmdr1*, *arps10*, *ap-2m* or *ubp1* and the degree to which mutant *K13* conferred ART resistance in vitro in our set of African or Asian strains (Supplementary file 4).”

4. The authors' group and other groups have shown that the lower expression levels of K13 is responsible for the ART resistance phenotype in earlier studies.It would be interesting to check whether the strains which show higher survival rates after introduction of K13 mutations also have a lower K13 expression.

We have now completed a series of Western blot studies with the 3D7 parasite lines, prepared and assayed on three separate occasions. Results show a reduction in K13 basal expression levels in synchronized ring-stage parasites expressing R561H, M579I or C580Y. A representative Western blot is now included in Figure 2—figure supplement 1. Quantification across experiments yielded an expression level of ~70% in the K13 mutant lines relative to K13 WT. These three variants showed similar levels of resistance in the RSA. These data are consistent with our other results obtained with NF54 and Cam3.II lines, reported in Gnadig et al., 2020, PLoS Pathogens and Mok et al., 2021, Nature Comm. Given the complexity of performing these studies with large volumes of highly synchronized early ring-stage parasites, and our need to also complete multiple fitness assays, we feel it best to extend these studies to other genetic backgrounds and report these data at a future date. Our new data are described above in the response to Comment #3 from the Editors.

5. In figure 5, it is shown that Dd2 eGFP parasite had considerable fitness costs, please provide a reason. Also, the data might be better presented as normalized against Dd2WT (using Dd2 eGFP as the reference).

Our Dd2^eGFP^ line has this expression cassette integrated into the *cg6* locus using the attB×attP recombination system (Nkrumah et al., 2006, Nature Methods). This locus also expresses both the human *dhfr* and *blasticidin S-deaminase* selectable markers. We speculate that the fitness cost observed with Dd2^eGFP^ parasites is a result of high-level GFP expression, driven by a calmodulin promoter sequence, and the genomic integration of the two selectable marker cassettes.

Our data contained herein, which show a reduced rate of Dd2^eGFP^ expansion compared with wild-type Dd2, are consistent with our prior fitness data reported in Ross et al., (2018, Nature Comm) and Dhingra et al., (2019, Lancet Infect Dis). A reference to these earlier studies is now included in our revised manuscript as follows:

Methods lines 580-584: “This reporter line uses a *calmodulin* promoter sequence to express high levels of GFP and includes human *dhfr* and *blasticidin S-deaminase* expression cassettes, and was earlier reported to have a reduced rate of growth relative to parental non-recombinant Dd2 (Ross et al., 2018; Dhingra et al., 2019).”

We have also revised our Figure 5 to show fitness data relative to Dd2^WT^, as listed above in our response to Reviewer 1 Comment #2.

6. The mdr2 and/or ferrodoxin mutations: given that these mutations are suspected to confer better fitness to the parasites, it makes sense to check the effect of fd D193Y and mdr2 T484I on parasite fitness.

We thank the Reviewer for this important suggestion. These experiments have now been completed and are described in our response to Comment #3 from the Editors.

7. In western Cambodia, WT allele percentage in 2001-2002 had already fallen to 56%, but ACT failure was detected much later. That's interesting, may be the authors can discuss this further.

We thank the Reviewer for this astute observation. The identification in 2001-02 of mutant K13 in 45 of 103 isolates (43%), of which half were C580Y, suggests that ART resistance was already widespread several years prior to the initial clinical reports (Noedl et al., 2008, NEJM; Dondorp et al., 2009, NEJM). These samples came from an earlier report (Ariey et al., 2014, Nature) that already had documented the high prevalence of mutant K13 in this 2001-02 time period. This was previously discussed in our Results section. Our revised text also now refers to this as follows:

Discussion lines 351-353: “Interestingly, this mutation was already at high prevalence in western Cambodia several years before the first published reports of delayed parasite clearance in ART-treated patients (Noedl et al., 2008; Dondorp et al., 2009; Ariey et al., 2014).”

8. Page 15: subtitle "Strain-dependent genetic..." needs to be revised.

Thank you. Our revised manuscript now reads:

Results lines 270-271: “Strain-dependent genetic background differences significantly impact RSA survival rates in culture-adapted Thai isolates”.

9. Page 16: line 6: better to clarify as "we applied CRISPR/Cas9 editing to revert the fd D193Y and mdr2 T484I mutations to their wild type in the…strains"

Our revised text now reads:

Results lines 291-293: “To directly test their role, we applied CRISPR/Cas9 editing (Supplementary file 6) to revert the fd D193Y and mdr2 T484I mutations to the wild-type sequences in the Cambodian strains RF7^C580Y^ and Cam3.II^C580Y^, which both express K13 C580Y.”

10. Given the extent of the study, the title is misleading and should be revised to reflect the large amount of data concerning the SE Asian parasites and mutations.

Our revised title (lines 1-2) is now: “*Plasmodium falciparum* K13 mutations in Africa and Asia impact artemisinin resistance and parasite fitness”.

Reviewer #3 (Recommendations for the authors):This manuscript by Stokes et al., combines a number of different studies on different parasites of different origins. It appears to be work put together from different labs conducted in some cases with different methods.

We would like to clarify that all gene editing and RSA studies were performed in the Fidock lab at Columbia University. Editing was performed using *k13*-specific zinc-finger nucleases, or more recently CRISPR/Cas9 editing, depending on when the transfections were performed. Our lab also performed all the fitness assays, with protocols customized depending on the parasite strain. K13 genotyping was performed by Didier Menard and colleagues at the Pasteur Institutes in Paris or Phnom Penh. Other coauthors provided isolates or parasite DNA samples from field sites.

The authors have compiled genotype data from isolates from 11 different countries in Africa. The authors note the continuing dominance (99%)of wild-type K13 and the presence of the K13 R561H variant in Rwanda. The Abstract suggests that 3,299 isolates were genotyped as part of the study. However, it seems that 2,261 of these genotype datasets have been reported previously, including the 927 parasite samples from Rwanda. The authors should make this clear in the text.

We have now clarified this in the Results section, as described above in our response to Reviewer 2 Comment #1.

It is notable that the Rwandan samples were collected from 2012 to 2015. As the authors point out more recent studies suggest that the prevalence of the K13 R561H variant remains relatively low. The authors should comment on the implications of this finding for spread of this variant.

Our revised manuscript now refers to two recent studies from Rwanda showing the presence of R561H in 28/218 (13%) and 8/66 (12%) field samples (Uwimana et al., 2021, Lancet Infect Dis; Bergman et al., 2021, Emerging Infect Dis). These studies examined samples from 2018 and 2019, respectively. These results compare with 20/927 (2%) of Rwandan R561H mutants observed in our samples collected from 2012 to 2015. These findings suggest an increasing prevalence of this K13 variant over time. Of note, the report by Uwimana et al., also observed a statistically significant association between R561H and day 3 positivity in patients having received artemether-lumefantrine. These updated data are now included in our revised manuscript as follows:

Introduction lines 95-104: “We include the K13 R561H mutation, earlier associated with delayed parasite clearance in SE Asia (Ashley et al., 2014; Phyo et al., 2016), and very recently identified at up to 13% prevalence in certain districts in Rwanda (Uwimana et al., 2020; Bergmann et al., 2021; Uwimana et al., 2021). […] Nonetheless, our data highlight the threat of the R561H mutation emerging in Rwanda, which confers elevated RSA resistance and minimal fitness cost in African 3D7 parasites.”

Discussion lines 342-348: “In contrast, we observed no evident fitness cost in 3D7 parasites expressing the R561H variant, which might help contribute to its increasing prevalence in Rwanda. […] These recent data heighten the concern that mutant K13 might be taking hold in certain areas in in Africa where it can begin to compromise ACT efficacy.”

The authors undertook further studies of three K13 mutations, R561H, C580Y and M579I. The M579I mutation was identified in a *P. falciparum*-infected individual in Equatorial Guinea who displayed delayed parasite clearance following ACT treatment. It is not clear why this variant was chosen for further study as it does not seem to have been identified in any of the 3299 samples compiled for the comparative analysis of genotypes.

The M579I mutation was first identified in a *P. falciparum*-infected adult male upon his return to China from Equatorial Guinea, where he had been working for 20 months. We chose to study this mutation because the individual was day 3 positive for *P. falciparum* and ex vivo RSAs showed evidence of ART resistance. Until the very recent report of R561H being associated with delayed parasite clearance in Rwanda (Uwimana et al., 2021, Lancet Infect Dis), this was the only evidence associating mutant K13 with ART resistance in Africa. We also now refer to this mutation in our revised manuscript as follows:

Discussion lines 340-342: “An even great fitness cost was observed with the M579I mutation, earlier detected in an African infection with evidence of in vivo ART resistance (Lu et al., 2017) but which was notably absent in all 3,257 African samples reported herein.”

Introduction of R561H, C580Y and M579I K13 mutations into F32WT (African origin) parasites yielded almost no increase in RSA survival. This is surprising given that F32 is the strain that was used in the foundational long-term ART selection studies. The original study identified the M476I K13 mutation. Gene editing studies, performed by some of the authors of the current work, were used to validate this mutation in the F32 line (survival (<0.2% in F32-TEM to 1.7% in F32-TEMM476I)). The authors state that they developed a new CRISPR/Cas9-mediated K13 editing strategy to introduce the mutations. Did the authors validate the editing strategy used in this study (and the current clone of the F32 line) by introducing the M476I K13 mutation?

Our data in Figure 2 show that K13-mutant F32 parasites displayed RSA values that were consistently low across all three mutations tested (R561H, M579I and C580Y). This observation is consistent with our evidence that C580Y is a low to moderately resistant mutation, as shown with Africa and Asian strains (see Figures 2 and 5). These data were obtained using a new CRISPR/Cas9 editing strategy, as the Reviewer notes. Prior studies with other strains, obtained with zinc-finger nuclease-based gene editing, have also shown that C580Y confers a moderate degree of in vitro ART resistance when compared to other mutations such as the high-grade resistance mutation R539T (Straimer et al., 2015, Science; Straimer et al., 2017, mBio).

In Dd2 isogenic lines, we earlier showed that the M476I mutation conferred 9.8% survival compared with 4.1% survival in parasites expressing C580Y, corresponding to a 2.5 fold increase (Straimer et al., 2015, Science). This would be consistent with our finding here that our F32 parasites expressing C580Y had lower survival than F32 parasites expressing M476I. While we have not performed a head-to-head comparison of the two editing methods with the same mutation in the same parasite line, we note that both methods achieve gene editing without the introduction of any selectable markers or any modification beyond the targeted mutation. We therefore are confident that the gene editing strategies are not a reason behind the low values observed herein for F32. Instead, our evidence obtained with three distinct mutations suggest that K13-mutant F32 is intrinsically less resistant to ART compared with the other strains tested herein (3D7, UG659 and UG815). We clarify this point in our revised manuscript as follows:

Methods lines 478-480: “Both our customized zinc-finger nuclease and CRISPR/Cas9 approaches generated the desired amino acid substitutions without the genomic integration of any plasmid sequences or any additional amino acid changes in the K13 locus, and thus provide fully comparable data.”

The authors measured the fitness cost of the C580Y and M579I K13 mutations in parasites of African origin. These two mutations had fitness costs in all backgrounds. It is not clear why R561H was not included in this study, given that it is the only variant that appear to be persisting in an African setting. The authors conclude that the C580Y and M579I K13 mutations cause substantial fitness costs, and suggest that this may "counter-select against their dissemination in high-transmission settings". A comparison can be made with the C580Y mutation in the Dd2 (Indochina origin lab strain) background (Figure 5), but not the M579I mutation. And while there is data for R561H fitness cost in Dd2, there is no data for field strains of African and SEA backgrounds.

Please see our response to Comment #2 from the Editors, which summarizes our new data on fitness costs of R561H in the African strains 3D7 and F32.

Another feature of the data presented in Figure 5 and 6 is that different mutations have very different fitness costs in different backgrounds, which is not readily correlated with frequency at which the alleles are found in field parasites. Moreover, in agreement with the authors' previous studies, the levels of resistance offered by different mutations in highly dependent on the genetic background. Thus, while it remains a reasonable hypothesis, the data presented here are not sufficient to support the authors' conclusion that fitness costs determine whether K13 mutations persist in the African setting.

We agree with the Reviewer that fitness costs alone cannot explain differences in the prevalence of K13 mutants in different geographic regions, and our manuscript supports important roles for both the degree of resistance conferred by distinct K13 mutations and the contribution of the genetic background to the ART resistance. Our revised manuscript clarifies that fitness costs alone do not determine the persistence of K13 mutations in African settings, although we do propose that fitness costs are one factor. Several edits shown above refer to this. We also refer to this in our revised manuscript as follows:

Abstract lines 41-42: “These data underline the complex interplay between K13 mutations, parasite survival, growth and genetic background in contributing to the spread of ART resistance.”

Discussion lines 363-371: “Our studies into the impact of K13 mutations on in vitro growth in Asian Dd2 parasites provide evidence that the C580Y mutation generally exerts less of a fitness cost relative to other K13 variants, as measured in *K13*-edited parasites co-cultured with an eGFP reporter line[…] These might include specific genetic backgrounds that have favored the dissemination of C580Y parasites, possibly resulting in enhanced transmission potential (Witmer et al., 2020), or ACT use that favored the selection of partner drug resistance in these parasite backgrounds (van der Pluijm et al., 2019).”

The authors genotyped K13 in 3,327 parasite isolates from 4 provinces of Cambodia and report changes in distribution of K13 alleles. This is a useful graphical analysis of the increasing dominance of C580Y; and tells the same story as several other recent studies of isolates from this region. It would be useful to explain the overlap (or differences) from sample sets used in other studies of this kind.

Please see our reply to Comment #1 from the Editors.

Of interest, gene edited lines expressing K13 P553L in Dd2 exhibited equivalent degrees of ART resistance and a similar low fitness cost as C580Y. But P553L remains a rare mutation in the field. This suggests that the dominance of the C580Y mutation in SEA is not completely explained by resistance level conferred and fitness cost suffered, as measured by currently available assays.

We fully agree. To date, there is no satisfactory explanation in the literature as to why C580Y has outcompeted other mutations. As discussed above, the relatively low fitness cost may be one contributing factor but clearly there are others, potentially including differences in transmission dynamics, genetic background effects, and selection pressure from the ACT partner drug piperaquine. Please see our response to Comment #3 from the Editors.

In summary, while an impressive amount of work from different labs is presented in this manuscript, a systematic analysis of the key R561H Rwandan mutations in well-validated field parasites of Asian backgrounds is lacking – making it difficult to draw firm conclusions about the relationship between fitness cost and allele persistence.

As mentioned above, our revised manuscript now includes fitness data from two African strains, where R561H poses the greatest threat. Our manuscript already includes data from the Asian Dd2 strain and those data are discussed in the context of the prior study by Nair et al., on R561H in one Thai isolate, as discussed in our reply to Comment #2 from the Editors.